# Engineered cross-feeding creates inter- and intra-species synthetic yeast communities with enhanced bioproduction

Young-Kyoung Park [1,2], Huadong Peng [1,3], Piotr Hapeta[1], Lara Sellés Vidal [1] & Rodrigo Ledesma-Amaro [1] ✉

Microorganisms can be engineered to sustainably produce a variety of products including fuels, pharmaceuticals, materials, and food. However, highly engineered strains often result in low production yield, due to undesired effects such as metabolic burden and the toxicity of intermediates. Drawing inspiration from natural ecosystems, the construction of a synthetic community with division of labor can offer advantages for bioproduction. This approach involves dividing specific tasks among community members, thereby enhancing the functionality of each member. In this study, we identify six pairs out of fifteen composed of six auxotrophs of *Yarrowia lipolytica* that spontaneously form robust syntrophic and synergistic communities. We characterize the stability and growth dynamics of these communities. Furthermore, we validate the existence of syntrophic interactions between two yeast species, *Y. lipolytica* and *Saccharomyces cerevisiae*, and find a strain combination, *Δtrp2* and *Δtrp4*, forming a stable syntrophic community between two species. Subsequently, we introduce a 3-hydroxypropionic acid (3-HP) biosynthesis pathway into the syntrophic community by dividing the pathway among different strains. Our results demonstrate improved production of 3-HP in both intra- and interspecies communities compared to monocultures. Our results show the stable formation of synthetic syntrophic communities, and their potential in improving bioproduction processes.

The advances in synthetic biology and metabolic engineering have led to improved biotechnology processes using microorganisms for the production of food, pharmaceuticals, biofuels, and biomaterials. Despite methodological advances in our capacities to improve microbial strains, some commonly found challenges remain, including metabolic burden due to the high level of pathway engineering, cofactor imbalance, or toxicity of intermediates and/or final products. To overcome the drawbacks of engineering single chassis strain, the establishment of synthetic microbial communities by engineering multiple strains that cooperate during the bioprocess has been

proposed[1–3]. By dividing the labor among multiple strains, synthetic communities are able to improve the functionality of each member, reduce metabolic burden and engineering complexity, and accomplish high efficiency of production as found in natural consortia[1,2,4].

In natural communities, there are various cellular interactions that determine the dynamics of the consortia, such as competition, commensalism, mutualism, or neutralism[5]. When it comes to synthetic consortia, designing a proper interaction between members are crucial for constructing a stable and robust synthetic community[2,3]. A type of mutualistic interaction, cross-feeding or syntrophy, requires each

[1]Department of Bioengineering and Centre for Synthetic Biology, Imperial College London, London, UK. [2]Université Paris-Saclay, INRAE, AgroParisTech, Micalis Institute, Jouy-en-Josas, France. [3]Australian Institute of Bioengineering and Nanotechnology, The University of Queensland, Brisbane, Queensland, Australia. ✉e-mail: r.ledesma-amaro@imperial.ac.uk

population that relies on each other for survival, which can provide stable coexistence by tying together the members in the community[5,6]. One way to achieve cross-feeding is by using co-auxotrophic strains that exchange essential amino acids to allow each other to grow[7,8].

It has been generally regarded that yeast co-cultures were not as effective as bacterial ones in forming co-auxotrophic communities, except for strains engineered to produce higher amount of amino acids[8]. Recently, we performed high-throughput screening of syntrophic interactions in the model yeast *Saccharomyces cerevisiae* by using yeast knockout library[9,10]. From this study, 49 pairwise auxotroph combinations which is 3.6% of tested pairs were identified to spontaneously form syntrophic communities and some of them were tested for division of labor, leading to improved bioproduction[10]. This finding suggests that cross-feeding-based communities could be formed in other yeast species, including those with high industrial potential.

*Yarrowia lipolytica* has been gaining interests as a host strain for bioproduction of chemicals, fuels, foods, and pharmaceuticals from both academia and industry[11,12]. Advantageous industrial features of this yeast include robustness, stress tolerance, being amenable by synthetic biology tools, and high cell density cultivation. Most research using *Y. lipolytica* have focused on engineering in a single strain. The studies on microbial communities using this yeast are so far limited. There are few studies of co-culture using *Y. lipolytica* with other species for bioremediation or feedstock utilization with the modulation of inoculation ratio or time among community members[13–20]. A study has explored division of labor for bioproduction of amorphadiene with *Y. lipoltyica* strains. A modular co-culture dividing the pathway for boosting precursor pools and amorphadiene synthesis resulted in the improved titers[21]. These works highlight the increasing interest in creating communities of *Y. lipolytica*. However, tools for controlling population dynamics, such as cross-feeding[9], to maximize robustness and efficiency have not yet been developed in *Y. lipolytica*.

In this study, we explored the creation of syntrophic communities of *Y. lipolytica* using auxotrophic strains and identified pairs exhibiting synergistic growths, which were further characterized. The *Y. lipolytica* auxotrophic strains were also evaluated for establishing the interspecies syntrophic growth with *S. cerevisiae* auxotrophs. We finally developed a division of labor strategy for the production of a bioplastic precursor, 3-hydroxypropionic acid, employing syntrophic intraspecies and interspecies communities, which resulted in increased bioproduction.

## Results

### Establishing synthetic *Y. lipolytica* communities by engineering cross-feeding behaviors

To evaluate if auxotrophs of *Y. lipolytica* could form syntrophic communities by exchanging essential metabolites, we constructed the strains Δlys5, Δtrp2, Δtrp4, Δmet5, Δura3 and Δleu2, auxotrophic for lysine, tryptophan, methionine, uracil, and leucine.

The growth of 15 paired combinations from these six auxotrophs was tested at a 1:1 inoculation ratio in YNBD media without amino acid supplementation. The observed growth of the tested combinations can be grouped into three categories according to their maximal $OD_{600}$ during the cultivation (Fig. 1a, Supplementary Table 1, Supplementary Figs. 1, 2): high ($OD_{600} \geq 0.55$): Δura3-Δtrp4, Δura3-Δmet5, Δleu2-Δtrp4, Δlys5-Δtrp4, and Δtrp4-Δmet5; moderate ($0.32 \leq OD_{600} < 0.55$): Δura3-Δlys5, Δura3-Δtrp2, Δleu2-Δtrp2, Δlys5-Δtrp2, Δlys5-Δmet5, and Δtrp2-Δtrp4; and low ($OD_{600} < 0.32$): Δura3-Δleu2, Δleu2-Δlys5, Δleu2-Δmet5, and Δtrp2-Δmet5. Among the high-growth combinations, three pairs (Δleu2-Δtrp4, Δlys5-Δtrp4, and Δtrp4-Δmet5) showed a constant increase in growth, while the other two pairs (Δura3-Δtrp4 and Δura3-Δmet5) exhibited an exponential growth after a certain time of lag phase (40 and 20 hours, respectively) (Fig. 1b, Supplementary Fig. 1). Positive correlation between growth and

glucose consumption depending on the auxotroph pairs was observed (Supplementary Fig. 3). The shortest lag phase, 12 hours, was found in the combination of Δtrp2 and Δtrp4, and the stationary phase was reached at 36 hours of cultivation. While the final $OD_{600}$ reached by the Δtrp2 and Δtrp4 did not rank amongst the top, it exhibited the highest growth rate compared to other combinations (Supplementary Table 1). The prolonged lag phase in some of the synthetic communities could be originated by the needs of each population to adapt their metabolism to export metabolites, which is required by its partner, and/or to import metabolites secreted from the partner[10].

### Characterization of growth dynamics of synthetic cross-feeding communities

We characterized population dynamics of three selected pairs (Δura3-Δtrp4, Δtrp4-Δmet5, and Δtrp2-Δtrp4) by varying the inoculation ratios from 10:1 to 1:10 (Fig. 2). Changes in inoculation ratios exhibited considerable differences in growth patterns, which suggest that certain population ratios favor syntrophic growth. In the pair of Δura3-Δtrp4, the inoculation ratios of 10:1 and 5:1 showed a shorter lag phase than other ratios, suggesting the importance of having more Δura3 cells at the beginning of the culture (Fig. 2a). Despite the shorter lag phase for these ratios, all inoculation ratios reached a similar final $OD_{600}$ at 120 hours. In addition, regardless of the initial ratio, the population tended to stabilize at the end of the stationary phase, maintaining a ratio of Δura3:Δtrp4 between 1:1.2 and 1:1.8 (Fig. 2d, g, Supplementary Fig. 5). In the case of Δmet5-Δtrp4 pair, the coculture with initial ratios of 1:1, 1:5 and 1:10 started growing earlier than those with 10:1 and 5:1 (Fig. 2b). At ratios of 10:1 and 5:1, the cocultures showed a mild growth until 48 hours followed by exponential phase. The final $OD_{600}$ was correlated with the inoculation ratio from 1:10 to 10:1 which also corresponded to glucose consumption (Supplementary Fig. 4). At the stationary phase, the population ratio was stabilized in all cases (Δmet5:Δtrp4 between 1:1.0 and 1:1.9 (Fig. 2e, h, Supplementary Fig. 6). The pair of Δtrp2-Δtrp4 showed a faster growth at all inoculation ratios compared to other combination tested (Fig. 2c). The ratio 1:10 showed the shortest lag followed by 1:5 and 1:1, while the ratios of 5:1 and 10:1 resulted in a longer lag phase and lower final $OD_{600}$. Since the mutations Δtrp2 and Δtrp4 are both mapped in the same tryptophan synthesis pathway, the growth between these two auxotrophs is achieved by the exchange of intermediates, which are known to be anthranilate and indole/tryptophan in *S. cerevisiae*[10]. In *S. cerevisiae*, the coculture of Δtrp2-Δtrp4 was naturally highly enriched in one population, Δtrp2 cells, after inoculating at 1:1 ratio. Similarly, a majority of Δtrp2 cells were found in the *Y. lipolytica* coculture at 10:1 and 5:1 inoculation ratios, which showed a slower and lower growth compared to other inoculation ratios (Fig. 2f, Supplementary Fig. 7). However, in *Y. lipolytica*, the ratio resulting in better growth exhibited different population dynamics, the ratio of Δtrp2:Δtrp4 at the stationary phase was 1:1.5 from the inoculation ratio of 1:5 and 1:10 (Fig. 2i). These considerable distinct population dynamics between two species suggest differences in metabolite exchange rates or mechanisms between *S. cerevisiae* and *Y. lipolytica*.

### Establishing cross-feeding communities between two yeast species, *Y. lipolytica* and *S. cerevisiae*

After demonstrating the formation of stable cross-feeding co-cultures between two *Y. lipolytica* auxotrophs, we decided to test whether the syntrophic growth could be established between the two species, *Y. lipolytica* and *S. cerevisiae*. We selected the *Y. lipolytica* auxotrophs described above (YLΔtrp2, YLΔtrp4, YLΔmet5, and YLΔlys5) and the *S. cerevisiae* ones based on our previous study (SCΔtrp2, SCΔtrp4, SCΔmet5, and SCΔlys5)[10]. Three pairs (YLΔtrp2-SCΔtrp4, SCΔtrp2-YLΔtrp4, and SCΔmet5-YLΔtrp4) showed syntrophic growth in the interspecies coculture (Fig. 3a, Supplementary Fig. 8). The growth

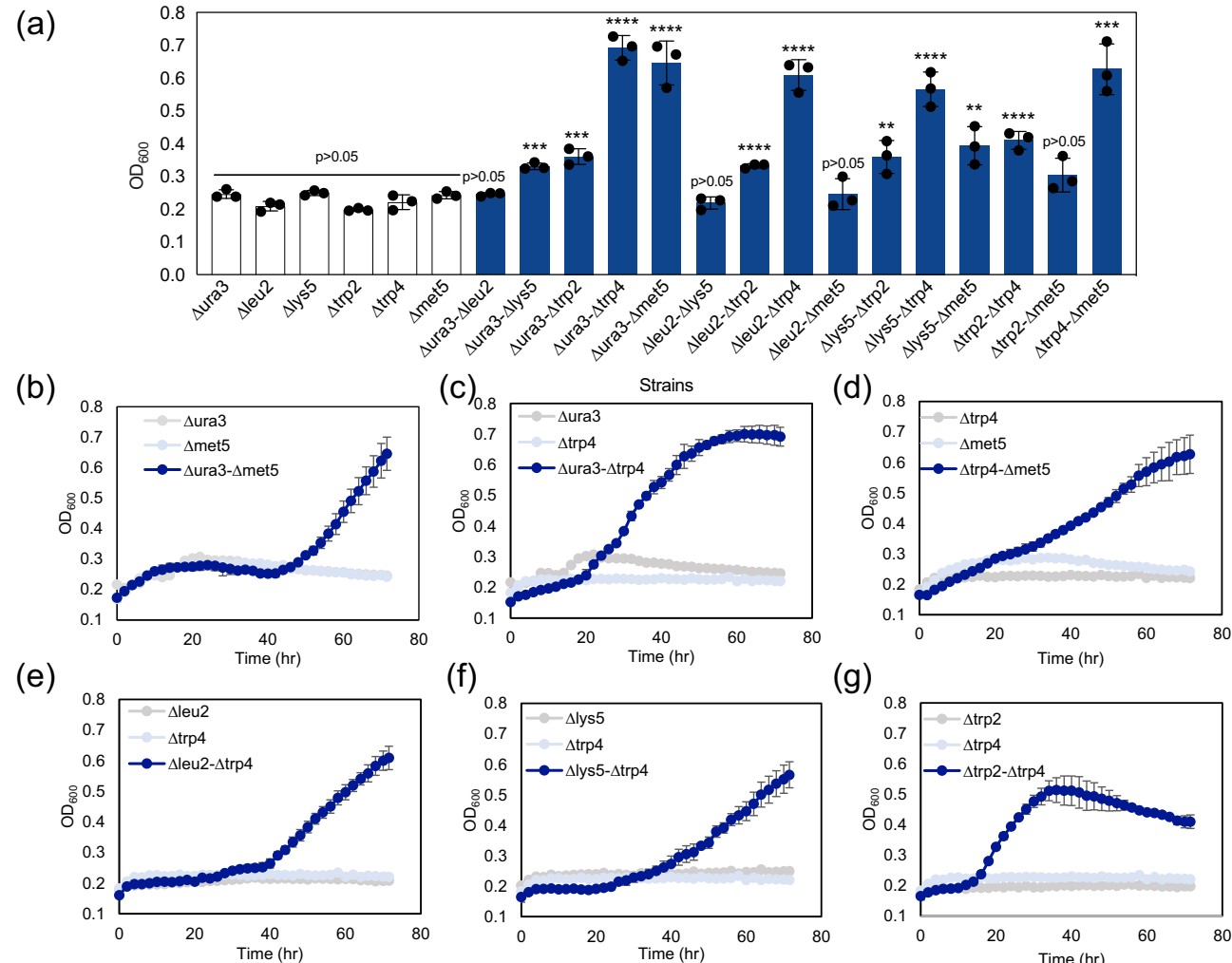

**Fig. 1 | Syntrophic cocultures of *Y. lipolytica* auxotroph strains. a** $OD_{600}$ values of monocultures (a single auxotroph, empty bars) and cocultures (a pair of two auxotroph with 1:1 inoculation ratio, blue bar) at 72 hours of cultivation. **b**–**g** Growth profile of successful cocultures. Auxotroph monocultures (negative controls, gray and light blue) were tracked along with the corresponding coculture (navy). $N = 3$ biologically independent samples and data are presented as mean ± standard deviation. One-way ANOVA, followed by Bonferroni's multiple comparisons test with 95% confidence intervals was performed using GraphPad Prism 9.5.0 software and $p$ values are indicated as asterisks in the graph (*: $p < 0.05$, **: $p < 0.005$, ***: $p < 0.0005$, ****: $p < 0.0001$). Source data are provided as a Source Data file.

dynamics differed depending on the auxotrophies and species involved. For example, the pair of YLΔ*met5*-SCΔ*trp4* did not grow while SCΔ*met5*-YLΔ*trp4* showed higher $OD_{600}$ than the coculture of YLΔ*met5*-YLΔ*trp4*. This might be due to the different metabolite exchange rates among species. As SCΔ*trp2*-YLΔ*trp4* pair showed higher growth compared to other auxotrophic pairs, we further characterized the population dynamics of SCΔ*trp2*-YLΔ*trp4* and YLΔ*trp2*-YLΔ*trp4* by inoculating different ratios (Fig. 3b–e, Supplementary Figs. 9 and 10). Both cocultures showed better growth at 1:1, 1:5, and 1:10 initial ratios. However, the inoculation ratios of 10:1 and 5:1 in SCΔ*trp2*-YLΔ*trp4* failed to grow, which could indicate that the exchange of the intermediate (anthranilate) was not enough for SCΔ*trp2* to grow in these conditions. The populations of SCΔ*trp2*-YLΔ*trp4* stabilized between the ratios of 1:0.9 and 1:1.5 at the stationary phase, which differs from the skewed population distribution of the SCΔ*trp2*-SCΔ*trp4* coculture. To validate whether different cultivation conditions affect the syntrophic growth of SCΔ*trp2*-YLΔ*trp4*, especially regarding a potential influence of the Crabtree effect, co-cultures of SCΔ*trp2*-YLΔ*trp4* with different glucose concentrations (20 and 100 g/L) and aeration condition (aerobic and semi-anaerobic) were performed (Supplementary Fig. 11). At 20 g/L of glucose, the coculture SCΔ*trp2* : YLΔ*trp4* showed

growth at 1:1 ratio in semi-anaerobic conditions, while no growth was observed in aerobic conditions. We observed a higher production of ethanol in the 1:1 ratio than in the 1:10 ratio, suggesting the Crabtree effect helped the growth of SCΔ*trp2*. At a higher glucose concentration (100 g/L), in both aerobic and semi-anaerobic conditions, we generally observed higher growth when there was a higher presence of the Δ*trp4* strain. In the SC-YL co-culture, we observed the Crabtree effect, as reflected by the ethanol production (100 g/L glucose, semi-anaerobic condition) that seemed to come from SCΔ*trp2*. As expected, a negligible amount of ethanol was observed in YL-YL co-culture in the same condition.

## Division of labor in cross-feeding communities improves bioproduction

Splitting metabolic pathways between strains in communities (division of labor) can be effective for bioproduction as it can reduce metabolic burden and avoid bottlenecks or toxic intermediates[2]. As a proof of concept, we aimed to synthesize a value-added molecule by splitting the biosynthesis pathway into a cross-feeding community. We selected the Δ*trp2*-Δ*trp4* pair because of its shorter lag phase and more rapid growth (Fig. 1).

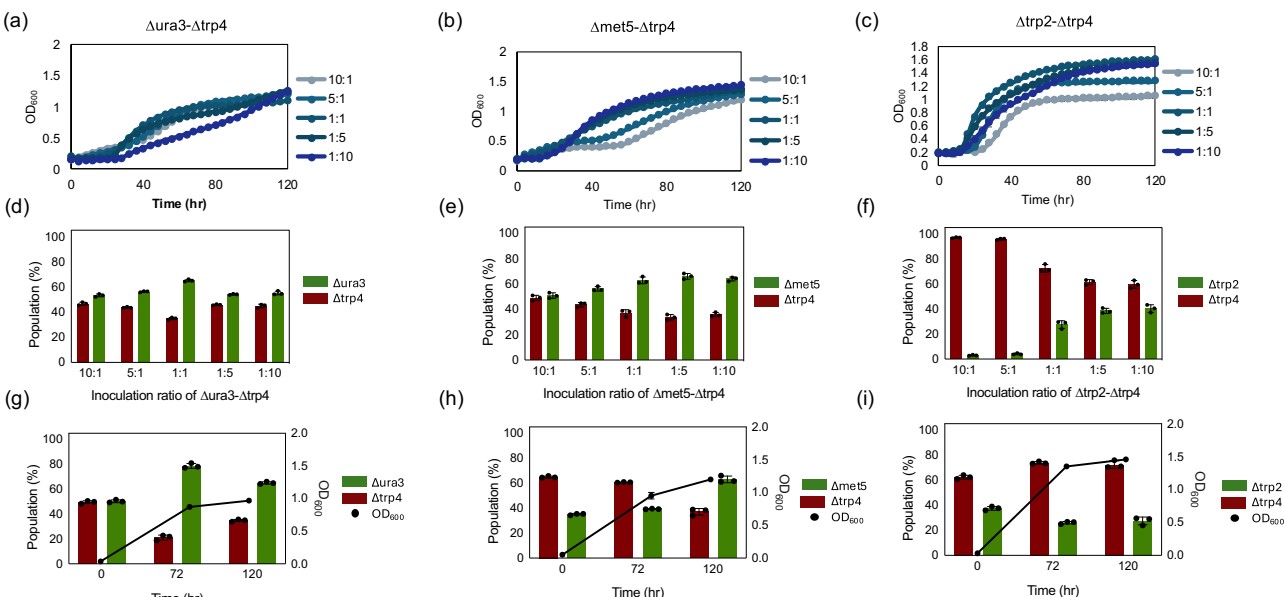

**Fig. 2 | Growth and population of coculture between auxotroph strains with different inoculation ratio.** Growth curve from the coculture of (**a**) *Δtrp4-Δura3*, (**b**) *Δtrp4-Δmet5*, and (**c**) *Δtrp2-Δtrp4*. Population ratios reached at stationary phase depending on inoculation ratio in the coculture of (**d**) *Δtrp4-Δura3*, (**e**) *Δtrp4-Δmet5*, and (**f**) *Δtrp2-Δtrp4*. Population dynamics at the 1:1 ratio from the combination of (**g**) *Δtrp4-Δura3*, (**h**) *Δtrp4-Δmet5*, and (**i**) *Δtrp2-Δtrp4*. Populations were measured by flow cytometry. *N* = 3 biologically independent samples and data are presented as mean ± standard deviation. Source data are provided as a Source Data file.

3-Hydroxypropionic acid (3-HP, $C_3H_6O_3$) is a desired platform chemical with a wide range of applications, as identified by the US Department of Energy in 2004. It is a precursor of acrylic acid, 1, 3-propanediol, malonic acid, biodegradable polyesters, and other valuable chemicals[22,23]. In order to synthesize 3-HP, we selected the biosynthetic pathway through β-alanine and malonic semialdehyde, which has not yet been applied in *Y. lipolytica*. This pathway requires the expression of three enzymes, aspartate-1-decarboxylase (TcPAND from *Tribolium castaneum*), β-alanine-pyruvate aminotransferase (BcBAPAT from *Bacillus cereus*), and 3-hydroxypropanoate dehydrogenase (EcYDFG from *Escherichia coli*) (Fig. 4a)[23]. The 3-HP pathway was split into module P, expressing TcPAND, and module B, expressing BcBAPAT and EcYDFG. Each module is integrated into two distinct auxotrophic strains, thereby generating a community that relies on the transport of β-alanine from one strain to another for producing 3-HP (Fig. 4b).

First, we tested intraspecies *Y. lipolytica* communities with division of labor and compared them with two WT controls, monoculture and coculture (Fig. 4, Supplementary Figs. 12 and 15). The WT monoculture bears three enzymes without division of labor and the WT coculture is composed of two strains, one harboring module P (WT-P) and the other module B (WT-B), thereby implementing division of labor but without cross-feeding. The growth was comparable between the WT monoculture and the WT coculture (Fig. 4c). The production of 3-HP was two times lower in the WT coculture compared to the WT monoculture (Fig. 4d), suggesting that division of labor, without cross-feeding, was not beneficial. In the case of the cross-feeding communities, the growth varied depending on the inoculation ratio of the strains *Δtrp2* with module B (*Δtrp2*-B) and *Δtrp4* with module P (*Δtrp4*-P) (Fig. 4c), which is consistent with the result shown in Fig. 3c. When the initial ratio was 1:10 (*Δtrp2*-B:*Δtrp4*-P), the synthetic community reached a similar OD600 as the WT monoculture after 48 hours of cultivation (Fig. 4c). However, the growth of coculture with initial ratios of 1:1 and 10:1 resulted in lower growth. The production of metabolites varied significantly with the inoculation ratio. Coculture at the ratio of 1:10 showed a comparable 3-HP production (0.26 mM) to one from the WT monoculture. The coculture of *Δtrp2*-B and *Δtrp4*-P

with an initial ratio of 10:1 reached a production of 4.67 mM of 3-HP, which is 19.3 times higher than the WT monoculture. Instead, WT monoculture produced higher citrate than co-culture (Supplementary Figs. 15 and 16). The higher ratio of module B in the communities showed higher 3-HP production, suggesting that the conversion of β-alanine to 3-HP is more important than the one from L-aspartate to β-alanine for higher production of 3-HP.

As cross-feeding communities were successfully established between *Y. lipolytica* and *S. cerevisiae*, we then decided to study the division of labor within this interspecies community. The strain pairs YL*Δtrp2*-SC*Δtrp4* and SC*Δtrp2*-YL*Δtrp4*, each with different 3-HP synthesis modules, were cultured using different initial inoculation ratios (Fig. 4, Supplementary Fig. 13). Consistently with what was observed for the corresponding cocultures without 3-HP bioproduction modules (Fig. 3c), different growth and metabolite profiles were observed depending on the inoculation ratio (Supplementary Fig. 10). The production of 3-HP varied depending on the combination of species and modules used for 3-HP production. Higher 3-HP production was commonly obtained with the ratio of B:P = 10:1 which is consistent with the result of *Y. lipolytica* intraspecies communities. This also demonstrates an effective transport of β-alanine from the *Δtrp2*-P strain to the *Δtrp4*-B strain in the syntrophic community. The highest 3-HP production from interspecies communities was achieved at 10:1 ratio of SC*Δtrp2*-B:YL*Δtrp4*-P, reaching 4.50 mM which is 40.3 and 18.6 times higher than the one from the WT monoculture of *S. cerevisiae* and *Y. lipolytica*, respectively (Fig. 4d, Supplementary Fig. 14). Therefore, these results successfully demonstrated an improvement of 3-HP production through pathway split (and likely division of labor) in both types of synthetic cross-feeding communities, the intraspecies of *Y. lipolytica* and the interspecies of *Y. lipolytica* and *S. cerevisiae*.

## Discussion

In nature, many microorganisms are auxotrophs and therefore rely on external nutrients (including amino acids) for their growth[24]. This observation has inspired synthetic biologists to design synthetic communities using amino acid or nucleotide auxotrophic strains. The requirement on essential metabolites exchange promotes cooperative

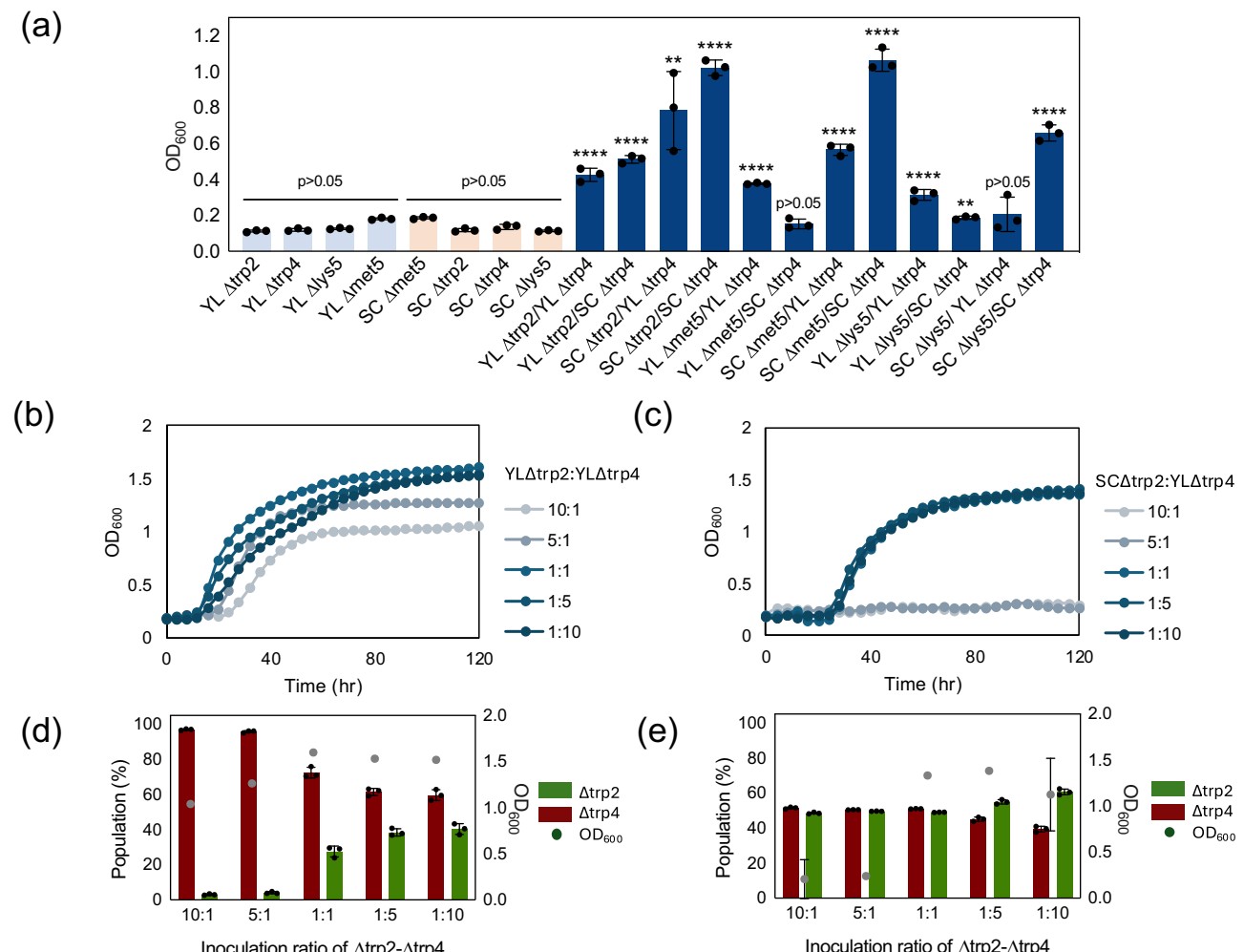

**Fig. 3 | Syntrophic coculture of *Y. lipolytica* (YL) and *S. cerevisiae* (SC) auxotroph strains. a** OD$_{600}$ values of monocultures (a single auxotroph, light blue for *Y. lipolytica* and light orange for *S. cerevisiae*) and cocultures (pairs of two auxotroph with 1:1 inoculation ratio, blue) at 72 hours of cultivation. **b** Growth of *Y. lipolytica* coculture in the YL*Δtrp2*-YL*Δtrp4* combination. **c** Growth of *Y. lipolytica* and *S. cerevisiae* coculture in the SC*Δtrp2*-YL*Δtrp4* combination. **d** population of each strain at different inoculation ratio in YL*Δtrp2*-YL*Δtrp4* coculture at 120 hours of cultivation. **e** population of each strain at different inoculation ratio in SC*Δtrp2*-YL*Δtrp4* coculture at 120 hours of cultivation. *N* = 3 biologically independent samples and data are presented as mean ± standard deviation. One-way ANOVA, followed by Bonferroni's multiple comparisons test with 95% confidence intervals were performed using GraphPad Prism 9.5.0 software and *p* values are indicated as asterisks in the graph (*: $p < 0.05$, **: $p < 0.005$, ***: $p < 0.0005$, ****: $p < 0.0001$). Source data are provided as a Source Data file.

behaviors and beneficial interactions. Recent studies on synthetic communities often require a high level of engineering to maintain the stability of the coculture and control the population[25,26], which limit the applicability and universality of these methods. Auxotrophic-based cross-feeding offers a simpler alternative to creating stable communities. However, identifying the adequate pairs of auxotrophs able to establish syntrophic interactions is challenging as metabolic costs and energy requirements for the synthesis of each amino acid or metabolite vary and their transport systems are not fully understood[27–29].

Here, we aimed to uncover spontaneous syntrophic communities in *Y. lipolytica*. Out of fifteen combinations involving six auxotroph strains, five exhibited robust syntrophic growth, and six demonstrated a slower but still discernible growth at an initial ratio of 1:1. Further investigation by modulating the initial inoculation ratio could potentially unveil additional auxotrophic pairs capable of establishing syntrophic communities. Generally, the success of syntrophic interaction is thought to be determined by the rates of import, export, and consumption of the involved metabolites, as the depletion of one of the metabolites before establishing the syntrophy can lead to the collapse of the community[30]. Therefore, pairs that failed to establish spontaneous syntrophic interactions might be attributed to low production

or a limited transport system of specific metabolites that need to be provided to the other member of the community. Engineering strains to overproduce specific metabolites through the regulation of feedback inhibition or the strengthening carbon flux towards their synthesis could be beneficial in promoting syntrophic interactions, as demonstrated independently in both *E. coli* and *S. cerevisiae*[8,9,31]. At a more fundamental level, it would be beneficial to study the transport mechanisms of metabolites, including amino acids in *Y. lipolytica*. Understanding the secretion or uptake of metabolites is pivotal in order to improve stable syntrophic interactions. Employing omics approaches, such as metagenomic sequencing[7] and exometabolomic analysis[32,33] could contribute to unravel some of these transport systems and better understand microbial cross-feeding within synthetic communities.

In this work, we also demonstrated spontaneous syntrophic growth between two yeast species, *Y. lipolytica* and *S. cerevisiae*. A pair of *Δtrp2*-*Δtrp4* demonstrated successful syntrophic interaction between two strains regardless of the combination of the auxotroph pairs and the species. In *S. cerevisiae* communities, *Δtrp2*-*Δtrp4* has been described to exhibit an extremely unbalanced population distribution, with one strain dominating the coculture (over 95% of

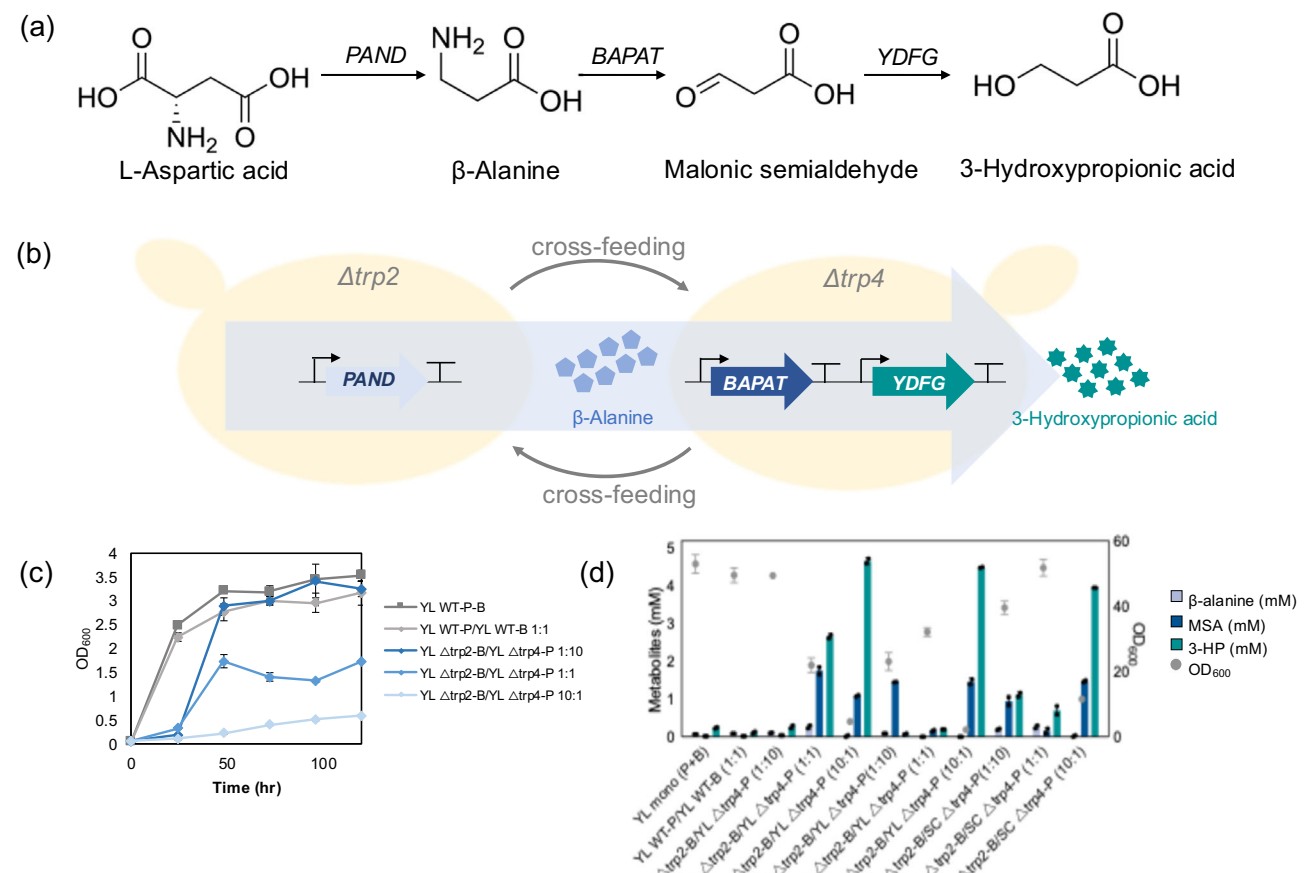

**Fig. 4 | Division of labor in syntrophic community for bioproduction of 3-hydroxypropionic acid. a** Biosynthetic pathway of 3-hydroxypropionic acid. **b** Strategy of division of labor in the synthetic community. **c** Growth of *Y. lipolytica* intraspecies syntrophic community. **d** Production of metabolites in the 3-HP synthetic pathway from the intraspecies and interspecies synthetic communities. $N = 2$ biologically independent samples and data are presented as mean ± standard deviation. Source data are provided as a Source Data file.

$\Delta trp2$)[10]. A similar trend was observed in the *Y. lipolytica* communities at inoculation ratios of 10:1 and 5:1, however, the population was more balanced ($\Delta trp2$:$\Delta trp4$ of 1:0.9-1.5) at the ratio of 1:5 and 1:10 (Fig. 2f, i). In the interspecies coculture of $\Delta trp2$-$\Delta trp4$, a balanced population was achieved in all tested strains, species, and ratios (Fig. 3e), highlighting the potential of interspecies syntrophic communities to provide an additional level of control. In specific inoculation ratios (10:1 and 1:1) of the SC$\Delta trp2$-YL$\Delta trp4$ coculture, growth failed to occur, suggesting an insufficient exchange of metabolites in this experimental condition. Similarly, the pair of YL$\Delta met5$-SC$\Delta trp4$ was unable to grow, while SC$\Delta met5$-YL$\Delta trp4$ grew. This observation might also be explained by different exchange rates of metabolites in different species[7,10]. Interdependent cocultures for bioproduction have so far mostly been explored using model microorganisms[1,34]. Our results suggest a broader applicability of syntrophic interactions beyond model microorganisms, paving the way for designing synthetic communities of non-conventional yeasts for bioproduction.

To study the effect of division of labor and cross-feeding in bioproduction by synthetic communities, we divided the biosynthetic pathway of 3-HP into two modules. The coculture of YL$\Delta trp2$-B and YL$\Delta trp4$-P, with an initial ratio of 10:1, produced 19.3 times higher 3-HP (4.67 mM) than the WT monoculture harboring the complete 3-HP synthetic pathway in a single strain. Notably, this synthetic pathway converting β-alanine into 3-HP was investigated in *Y. lipolytica* for the first time in this study. The growth and metabolite analysis (Supplementary Fig. 16) suggests that the higher 3-HP production found in the co-cultures originated from a higher availability of pyruvate, a

common precursor of 3-HP and citrate. This result underscores that the division of labor within a synthetic community can be used to validate undiscovered synthetic pathways, in addition to the traditional approach of embedding the entire pathway in a single strain. The production of 3-HP was further improved in the interspecies cross-feeding community, YL $\Delta trp2$-B and SC $\Delta trp4$-P at a 10:1 inoculation ratio, reaching 3.96 mM of 3-HP. This is slightly higher than the reported 0.35 g/L (3.88 mM) of 3-HP production in *Y. lipolytica* harboring the alternative pathway from malonyl-CoA[22].

When it comes to MSA production in *S. cerevisiae* communities, the coculture of $\Delta trp2$-B:$\Delta trp4$-P = 10:1 produced the highest MSA among different inoculation ratios but also outperformed the monoculture, which is consistent with the MSA production in a previous study of *S. cerevisiae* communities (Supplementary Table 4 and Supplementary Figs. 14 and 18)[10]. However, the production of MSA in *S. cerevisiae* co-culture at $\Delta trp2$-B:$\Delta trp4$-P = 1:1 and 1:10 was negligible, although it was higher than the monoculture in the previous study. This might be due to the different promoters used for expressing BAPAT in each study, additional gene expression (YDFG) in this study, and different cultivation scales.

In the case of 3-HP production, *S. cerevisiae* co-culture at specific inoculation ratio ($\Delta trp2$-B:$\Delta trp4$-P = 1:1) performed better than the *S. cerevisiae* monoculture (Supplementary Fig. 14, Supplementary Table 4). The level of total metabolites produced from the $\Delta trp2$-B strain (MSA and 3-HP) in the coculture of $\Delta trp2$-B:$\Delta trp4$-P = 10:1 is higher than the one from the monoculture (Supplementary Table 4 and Supplementary Fig. 14). In this study, we used the biosynthetic

**Table 1 | Plasmids and strains used in this study**

| Plasmid | | | Reference |
|---|---|---|---|
| RLA p603 | ZUS1.1-pTEF-RFP turbo-tLip2 | | This study |
| RLA p1503 | ZLS1.1-pTEF-RFP turbo-tLip2 | | This study |
| RLA p1506 | ZUS1.1-pTEF-hrGFP-tLip2 | | This study |
| RLA P1676 | pTDH3-*TcPAND*-tENO1-vLeu2 | | This study |
| RLA P1679 | pCCW12-*BcBAPAT*-tSSA1- pPGK1-*EcYDFG*-tADH1-vLeu2 | | This study |
| RLA P1683 | pTDH3-*TcPAND*-tENO1- pCCW12-*BcBAPAT*-tSSA1- pPGK1-*EcYDFG*-tADH1-vLeu2 | | This study |
| RLA p1475 | ZLS1.1-pTEF-hrGFP-tLip2 | | This study |
| RLA p2703 | pJET-*PAND* | | This study |
| RLA p2704 | pJET-*BAPAT* | | This study |
| RLA p2705 | pJET-*YDFG* | | This study |
| RLA p2706 | ZUS1.1-pTEF-*PAND*-tLip2 | | This study |
| RLA p2707 | ZUS1.1-pTEF-*BAPAT*-tLip2 | | This study |
| RLA p2708 | ZUS1.2-pTEF-*YDFG*-tLip2 | | This study |
| RLA p2709 | ZLA2.II- pTEF-BAPAT-tLip2-pTEF-*YDFG*-tLip2 | | This study |
| ***Yarrowia lipolytica***[a] | | Abbreviation | Reference |
| RLA S911 | po1d ZLS-pTEF-hrGFP-T3Lip2 | Δ*ura3*-GFP | This study |
| RLA S912 | po1d ZUS-pTEF-RFPturbo-T3Lip2 | Δ*leu2*-RFP | This study |
| RLA S1258 | po1d ZLS1.1-pTEF-RFPturbo-TLip2 | Δ*ura3*-RFP | This study |
| RLA S1260 | po1d ZUS1.1-pTEF-hrGFP-TLip2 | Δ*leu2*-GFP | This study |
| RLA S2543 | po1d Δ*lys5* ZLS1.1-pTEF-hrGFP-TLip2 + *URA3* | Δ*lys5*-GFP | This study |
| RLA S2544 | po1d Δ*lys5* ZUS1.1-pTEF-RFP turbo-TLip2 + *LEU2* | Δ*lys5*-RFP | This study |
| RLA S2547 | po1d Δ*trp2* ZLS1.1-pTEF-hrGFP-TLip2 + *URA3* | Δ*trp2*-GFP | This study |
| RLA S2548 | po1d Δ*trp2* ZUS1.1-pTEF-RFP turbo-TLip2 + *LEU2* | Δ*trp2*-RFP | This study |
| RLA S2551 | po1d Δ*trp4* ZLS1.1-pTEF-hrGFP-TLip2 + *URA3* | Δ*trp4*-GFP | This study |
| RLA S2552 | po1d Δ*trp4* ZUS1.1-pTEF-RFP turbo-TLip2 + *LEU2* | Δ*trp4*-RFP | This study |
| RLA S2555 | po1d Δ*met5* ZLS1.1-pTEF-hrGFP-TLip2 + *URA3* | Δ*met5*-GFP | This study |
| RLA S2556 | po1d Δ*met5* ZUS1.1-pTEF-RFP turbo-TLip2 + *LEU2* | Δ*met5*-RFP | This study |
| RLA S3427 | po1d ZUS1.1-pTEF-*PAND*-tLip2 + *LEU2* | *PAND* | This study |
| RLA S3428 | po1d ZLA2.II- pTEF-*BAPAT*-tLip2-pTEF-*YDFG*-tLip2 + *URA3* | *BAPAT* | This study |
| RLA S3429 | po1d ZUS1.1-pTEF-*PAND*-tLip2 + ZLA2.II- pTEF-*BAPAT*-tLip2-pTEF-*YDFG*-tLip2 | *PAND-BAPAT* | This study |
| RLA S3430 | po1d Δ*trp2* ZUS1.1-pTEF-*PAND*-tLip2 + *LEU2* | Δ*trp2-PAND* | This study |
| RLA S3431 | po1d Δ*trp2* ZLA2.II- pTEF-*BAPAT*-tLip2-pTEF-*YDFG*-tLip2 + *URA3* | Δ*trp2-BAPAT* | This study |
| RLA S3432 | po1d Δ*trp2* ZUS1.1-pTEF-*PAND*-tLip2 + ZLA2.II- pTEF-*BAPAT*-tLip2-pTEF-*YDFG*-tLip2 | Δ*trp2-PAND-BAPAT* | This study |
| RLA S3433 | po1d Δ*trp4* ZUS1.1-pTEF-*PAND*-tLip2 + *LEU2* | Δ*trp4-PAND* | This study |
| RLA S3434 | po1d Δ*trp4* ZLA2.II- pTEF-*BAPAT*-tLip2-pTEF-*YDFG*-tLip2 + *URA3* | Δ*trp4-BAPAT* | This study |
| RLA S3435 | po1d Δ*trp4* ZUS1.1-pTEF-*PAND*-tLip2 + ZLA2.II- pTEF-*BAPAT*-tLip2-pTEF-*YDFG*-tLip2 | Δ*trp4-PAND-BAPAT* | This study |
| ***Saccharomyces cerevisiae*** | | | |
| RLA S335 | BY4741 Δ*met5* pTDH3-mScarlet-tADH1-Leu + pHUM | Δ*met5*-mScarlet | Aulakh et al. [10] |
| RLA S337 | BY4741 Δ*trp2* pTDH3-mScarlet-tADH1-Leu + pHUM | Δ*trp2*-mScarlet | Aulakh et al. [10] |
| RLA S338 | BY4741 Δ*trp4* pTDH3-mTagBFP2-tADH1-Leu + pHUM | Δ*trp4*-BFP | Aulakh et al. [10] |
| RLA S722 | BY4741 Δ*lys5* pTDH3-mScarlet-tADH1-Leu + pHUM | Δ*lys12*-mScarlet | Aulakh et al. [10] |
| RLA S1628 | BY4741 pTDH3-*TcPAND*-tENO1+ pHUM | *PAND* | This study |
| RLA S1629 | BY4741 pCCW12-*BcBAPAT*-tSSA1- pPGK1-*EcYDFG*-tADH1-tLeu2 + pHUM + *LEU2* | *BAPAT* | This study |
| RLA S1585 | BY4741 pTDH3-*TcPAND*-tENO1- pCCW12-*BcBAPAT*-tSSA1, pPGK1-*EcYDFG*-tADH1-tLeu2, pHUM + pHUM + *LEU2* | *PAND-BAPAT* | This study |
| RLA S1604 | BY4741△*trp2* + pTDH3-*TcPAND*-tENO1 + pHM + *LYS21* | Δ*trp2-PAND* | This study |
| RLA S1605 | BY4741△*trp2* pCCW12-*BcBAPAT*-tSSA1- pPGK1-*EcYDFG*-tADH1-tLeu2 + pHM + *LYS21* | Δ*trp2-BAPAT* | This study |
| RLA S875 | BY4741 Δ*trp4* + pTDH3-*TcPAND*-tENO1 + pHLM | Δ*trp4-PAND* | This study |
| RLA S894 | BY4741 Δ*trp4* + pCCW12-*BcBAPAT*-tSSA1- pPGK1-*EcYDFG*-tADH1-tLeu2 + pHLM | Δ*trp4-BAPAT* | This study |

[a]Gene and species names are written in Italic font.

pathway of 3-HP as a proof of concept, but further modifications can lead to improve titers. Increased production is expected through additional engineering strategies such as promoter engineering, multi-copy integration, and precursor and/or cofactor supply. Overall, this work demonstrates that the combination of cross-feeding and inoculation ratio to control population dynamics in synthetic yeast communities with division of labor has the potential to improve the production of valuable molecules. It is worth noting that further research is required to understand the complex relationship between division of labor and bioproduction and fully correlate them both.

The strategy described here could be expanded to multiple organisms (and their combination) and compounds of interest.

In conclusion, we successfully demonstrated the establishment of a stable synthetic cross-feeding yeast community employing auxotrophs of *Y. lipolytica*, an yeast of high industrial interest. Synthetic communities of *Y. lipolytica* were characterized in terms of growth and population dynamics, considering different auxotrophic pairs and inoculation ratios. Our findings confirmed that specific auxotrophs can exchange metabolites with other members, facilitating spontaneous growth in both intraspecies (*Y. lipolytica*) and interspecies (*Y. lipolytica* and *S. cerevisiae*) communities. We further explored the division of labor and bioproduction of 3-HP within these syntrophic communities. Notably, we found a 3-HP production improvement by 19.3 and 18.6 times when labor was divided in intra- and interspecies communities compared to the *Y. lipolytica* monoculture, respectively. This study represents the first demonstration of a division of labor for biosynthetic heterologous pathway using syntrophic communities of *Y. lipolytica*. Our findings shed light on the potential of utilizing non-conventional microorganisms to form enhanced synthetic communities for bioproduction of various value-added molecules.

## Methods

### Strains and media

The *E. coli* strains DH5α and TOP10 were used for plasmid construction. *E. coli* strains were grown at 37 °C in Luria–Bertani (LB) medium (containing 1% tryptone, 0.5% yeast extract, and 1% sodium chloride) or on LB agar with appropriate antibiotics.

PCR amplifications were performed in a PCR ProFlex™ (Applied Biosystems) with Q5 High-Fidelity DNA Polymerase (New England Biolabs). PCR fragments were purified with a QIAgen Purification Kit (Qiagen). The plasmids used in this study were constructed by Golden Gate Assembly, as described in Yuzbashev et al.[35]. In brief, each component for GG assembly was cloned to Lv0 plasmid by using BsmBI. Lv1 plasmid containing the specific overhang for Lv2 plasmid was then constructed by assembling the Lv0 plasmid consisting of promoter, gene, and terminator using BsaI. Finally, the Lv2 plasmid containing two or three transcription units was constructed by using BsmBI. To verify the correct construction of plasmids, PCR with GoTaq DNA polymerases (Promega) and digestion by restriction enzyme (New England Biolabs) were carried out.

*Y. lipolytica* was routinely grown at 30 °C in YPD medium which consists of 1% yeast extract, 2% peptone, and 2% glucose, or yeast synthetic medium (YNBD) which includes 0.17% yeast nitrogen base without amino acids and ammonium sulfate, 0.5% ammonium chloride, 50 mM phosphate buffer ($KH_2PO_4$-$Na_2HPO_4$, pH6.8), and 2% glucose. To prepare the solid medium, 1.5% agar was added to the respective liquid medium. To complement auxotrophic processes, uracil, leucine, lysine, methionine, or tryptophan were added at a concentration of 0.1 g/L, as necessary.

To introduce gene expression cassettes into *Y. lipolytica*, NotI-linearized plasmids were transformed into competent cells by the lithium acetate/DTT method. The gene expression cassettes were randomly integrated into the genome of *Y. lipolytica*. Transformants were selected on YNBD media containing the appropriate amino acids for their specific genotype. Positive transformants were then confirmed by colony PCR with Phire Plant Direct PCR master mix (Thermo Fisher). Auxotroph strains were constructed by homologous recombination of promoter and terminator region of marker gene. The resulting auxotroph strains were verified on YNBD media with/without the corresponding amino acids. The removal of the selection marker was carried out via the Cre-LoxP system. The strains and plasmids used in this study are listed in Table 1. The primers used for cloning and verification are listed in Supplementary Table 2. The sequence of heterologous genes for 3-HP synthesis are listed in Supplementary Table 3.

### Growth and fluorescence analysis of yeast co-culture in 96 well plate

The yeast strains were initially cultured in YPD medium and cultivated overnight at 30 °C and 250 rpm. The cells were then washed three times with distilled water. Subsequently, the cells were inoculated into a 96-well plate containing 200 µl of YNBD media in triplicate. The initial $OD_{600}$ of culture, both monoculture and co-culture, was adjusted to 0.1. The inoculation ratios of the co-culture varied between 10:1, 5:1, 1:1, 1:5, and 1:10. The plate was incubated at 30 °C with continuous shaking for 120 hours. The growth and fluorescence of each strain were monitored using a Spark Tecan or Biotek instrument every 30 min using the following settings: $OD_{600}$, absorbance at 600 nm; GFP, excitation at 485 nm and emission at 535 nm; RFP, excitation at 560 nm and emission at 620 nm; mTAGBFP2, excitation at 400 nm and emission at 465 nm; and mScarlet-I, excitation at 560 nm and emission at 620 nm.

### Population analysis by Flow cytometry

Population of each member in the synthetic community was calculated by the different fluorescence of each strain. Cell fluorescence was measured by an Attune NxT Flow Cytometer (Thermo Scientific) with the following settings: FSC 100 V, SSC 355 V, BL1 345 V, YL2 510 V. Attune Cytometric software was used for data collection. Fluorescence data was collected from at least 10,000 cells for each experiment with three biological replicates.

### Growth analysis of yeast co-culture for 3-HP production in flask

The strains were initially cultured in YNBD medium with tryptophan and cultivated overnight at 30 °C and 250 rpm. The cells were then washed three times with distilled water. Subsequently, the cells were inoculated into the flask containing 10 ml of YNBD media at the initial OD of 0.1, for both monoculture and co-culture. The cells were incubated at 30 °C with 250 rpm for 120 hours. Samples were taken during cultivation to measure the $OD_{600}$ and quantify metabolites in the pathway of 3-HP. $OD_{600}$ values from flask samples were measured by using cuvettes in a UV/Visible spectrophotometer (Biochrom WPA Lightwave II) then normalized by using calibration curve (Supplementary Fig. 17) to compare $OD_{600}$ values between microplate reader and spectrophotometer measured in this study. Raw $OD_{600}$ data from each spectrophotometer are included in Supplementary Table 5.

### Quantification of metabolites

Metabolites including glucose, glycerol, citrate, and ethanol were analyzed by HPLC. The supernatant of cultures at each time point was diluted twenty times with distilled water before the analysis. The HPLC system was equipped with an Thermo Fisher UltiMate 3000 system and Aminex HPX-87H column (300 mm × 7.8 mm, Bio-RAD, USA) coupled to UV (210 nm) and RI detectors. The mobile phase used was 0.01 N $H_2SO_4$ with a flow rate of 0.6 mL/min and the column temperature was T = 35 °C. The raw data from HPLC were processed by Chromeleon software (Thermo Scientific). Concentration of metabolites was quantified by the calibration curve of each standard.

Metabolites in the 3-HP pathway, β-alanine, malonic semialdehyde, and 3-HP, were analyzed by LC-MS. The supernatant of cultures at each time point was diluted four times with 50% acetonitrile for the analysis. The LC-MS system was equipped with an Agilent 1290 Infinity LC system with an Agilent 6550 quadrupole time-of-flight mass spectrometer. An Agilent Poroshell 120 HILIC-Z, 2.1 × 100 mm, 1.9 µm, column was used at a temperature of 45 °C with a solvent flow rate of 0.25 ml min⁻¹. LC separation was performed with buffer A (10 mM ammonium formate in water) and buffer B (10 mM ammonium formate in water:ACN 10:90 (vol:vol)). After 0.5 min at 98% buffer B, the composition was changed to 5% buffer B over 2.5 min, then held at 5% buffer B for 1 min. Injection volume was 1 µl, and negative ion spectra were recorded over a mass range of 100–1000 m/z at a rate of

1 spectrum per second. All metabolites were qualified by the functional m/z values. β-Alanine and 3-HP were quantified by the calibration curve of the standards. Malonic semialdehyde was semi-quantified by the standard curves of β-alanine[10]. The raw data from LC-MS were processed by Agilent MassHunter Qualitative Analysis software (Supplementary Table 6).

## Statistics and reproducibility

All experimental data were analyzed using Microsoft Excel 365 and Prism 9.5.0 (GraphPad) software. The error bars in the Figs. 1, 2, and 3 correspond to the standard deviation from $N = 3$ biologically independent samples as described in figure legend. Statistical analyzes were conducted using one-way ANOVA, followed by Bonferroni's multiple comparisons test with 95% confidence intervals, and $p$ values are provided in the source data. The error bars in the Fig. 4 correspond to the standard deviation from $N = 2$ biologically independent samples as described in figure legend.

## Reporting summary

Further information on research design is available in the Nature Portfolio Reporting Summary linked to this article.

## Data availability

All data generated or analyzed in this study are included in the manuscript and its supplementary information. Source data are provided with this paper.

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

## Acknowledgements

R.L-A. received funding from BBSRC (BB/R01602X/1, BB/T013176/1, BB/T011408/1 - 19-ERACoBioTech- 33 SyCoLim, BB/X01911X/1, BB/Y008510/1 – Engineering Biology Hub for Microbial Foods), EPSRC (AI-4-EB BB/W013770/1, and EEBio Programme Grant EP/Y014073/), Yeast4Bio Cost Action 18229, European Research Council (ERC) (DEUSBIO - 949080), the Bio-based Industries Joint (PERFECOAT-101022370) under the European Union's Horizon 2020 research and innovation programme and the European Innovation Council (EIC) under grant agreement No. 101098826 (SKINDEV). Also Bezos Earth Fund for their support to the Bezos Centre for Sustainable Protein. Imperial College London UKRI Impact Acceleration Account (EPSRC –EP/X52556X/1, BBSRC -BB/X511055/1). Y.-K.P received funding from the Bio-based Industries Joint (PERFECOAT - 101022370) under the European Union's Horizon 2020 research and innovation programme. H.P. from the European Union's Horizon 2020 research and innovation programme under Marie Skłodowska-Curie grant agreement No. 899987. P.H. received funding from BBSRC BB/T013176/1. LSV received funding from BB/T011408/1 - 19-ERACoBioTech- 33 SyCoLim. Thanks to D. J. Bell for the analytical support from the SynbiCITE Innovation and Knowledge Centre at Imperial College London.

## Author contributions

Y.K.P. and R.L.A. conceived the project. Y.K.P. constructed plasmids and *Y. lipolytica* strains, characterized the strains, and analyzed the results. H.P. constructed *S. cerevisiae* strains. P.H. constructed plasmids. L.S.V. performed preliminary coculture experiments, statistics and edited figures. R.L.A. supervised the work. Y.K.P. drafted the manuscript. All authors read and edit the manuscript.

## Competing interests

The authors declare no competing interests.
