## [Transparent Peer Review file · Nature Communications]

Engineered cross-feeding creates inter- and intra-species synthetic yeast communities with enhanced bioproduction

Corresponding Author: Dr Rodrigo Amaro

Version 0:

Reviewer comments:

Reviewer #1

(Remarks to the Author)

The manuscript by Park et al shows that an engineered pathway split between different cells leads to increased production. The stability of the split system in terms of the co-existence is maintained by engineered obligate mutualism between the cell types (complementary auxotrophies). Interestingly, the engineered mutualism led to stable growth in intra- as well as inter-species systems consisting of two popular cell factories, *Saccharomyces cerevisiae* and/or *Yarrowia lipolytica*. The work is of interest and importance to the synthetic biology field and provides a proof-of-concept for using engineered communities for sustainable chemical production. However, the study in its current state is mostly observational and does not add much new insights beyond what has been shown before in *S. cerevisiae* communities (by the same lab; e.g. see Aulakh et al. Nature chemical biology 2023). Below are my main points and suggestions towards improving the study.

1. There is neither conceptual distinction nor mechanistic insights compared to Aulakh et al. 2023. This could be addressed by a more detailed, molecular analysis of the co-cultures, using, e.g. proteomics. The main question is what drives higher production in the split system? If it is division of labour, what are the costs and where do they stem from? Or are there other biochemical reasons.

2. The main difference to Aulakh et al 2023, regarding pathway, seems to be that in the present work, they have the complete pathway up to 3-hydroxypropionate whereas in Aulakh et al 2023, they only reported the addition of the BAPAT and PAND. However, in this work, they quantify the end product 3-hydroxypropionate instead of the intermediate malonic semialdehyde. There needs to be a detailed comparison with the Aulakh et al work in terms of observed yields, productivity etc and explanation of differences. For example, it seems that the final product was not quantified in the previous work while the pathway intermediates are not reported here.

3. With regards to the enhancement of bioproduction, the authors use the full 3-HP pathway rather than the same pathway up to malonic semialdehyde as they reported in Aulakh et al 2023. It is unclear why they chose different lengths of the same pathway for these two works as the auxotroph pairs and the experimental design are otherwise almost identical.

Their results indicate that when 3-HP is used instead of MSA to quantify bioproduction efficiency of this synthetic pathway, *Yarrowia lipolytica* co-cultures and *Y. lipolytica* - *S. cerevisiae* cocultures perform better than the *Y. lipolytica* monoculture while the *S. cerevisiae* monoculture performs better than the *S. cerevisiae* co-cultures. This contradicts the results of malonic semialdehyde production reported in their previous publication (Aulakh 2023), however, the authors do not discuss why this might be and how these results may be reconciled.

4. The authors state that the *S. cerevisiae* strains used in the present study bear the complete 3-HP pathway as reported in Borodina et al 2015 and give Aulakh 2023 as the reference for the construction of this strain. However, in Aulakh 2023, they reported only the insertion of BcBAPAT and TcPAND in their strains. For example, in Aulakh et al 2023, Supplementary Table 10, EcYDFG is not mentioned as being present in any strain. However, in Table 1 of the present work, the authors report that strain "RLA S1629" contains EcYDFG and the source as Aulakh et al 2023. It is therefore unclear whether the authors used the complete 3-HP pathway in their previous work (Aulakh 2023) and did not report the presence of the EcYDFG gene being inserted into the strains they created in that work or if the authors created new strains for the present

work which bear EcYDFG in addition to the BcBAPAT and TcPAND.

5. There is a 100-fold discrepancy between the OD600 values reported in various figures. For example, OD600 range in Figure 1 is up to 0.8, but the OD600 maximum in Figure 4 (c) is 60. Is this solely due to the difference in growth between 96-well cultures and flasks or have two different instruments been used to measure the optical density without calibration to a single axis? How these OD measurements relate to cell numbers? Is the cell density in flasks 75-times that in 96 well plates?

6. Based on the growth curve in Figure 4c, it seems that the readings for most of the exponential phase are missing.

7. The results of the metabolite quantification via LC-MS should be provided in a table form in the supplementary data. The mass-spectrometry analysis methods are unclear – are these targeted or untargeted measurements? From the methods it seems like untargeted analysis, which is odd given that only 3 metabolites are reported. Were standards used to validate these? The authors should also submit the raw mass spectrometry data to a community repository as is custom for mass spectrometry experiments.

8. Ince, the authors do not provide tables for their bioproduction capacity metabolite quantifications, one can only infer from the Line 261 “4.50mM which is 40.3 times higher than WT *S. cerevisiae* 18.6 times higher than WT *Y. lipolytica*” that the titre in WT *Sc* is ~0.11mM and WT *Yl* is ~0.24mM. This suggests that *Yl* is a better host for the 3-HP production than *Sc*. Why is this?

9. Minor issues:

Most figures do not show actual data points, which should be included in all cases

Figure 3a : why is there a > sign above the monocultures ?

Line 76 - community should be communities
“could be originated by the needs of each population”

149 - “tryptophane”- extra “e”

Reviewer #2

(Remarks to the Author)

This manuscript by Park et al. elucidates the genetic backgrounds of yeast strains, specifically *Yarrowia lipolytica* and *Saccharomyces cerevisiae*, facilitating both intra- and inter-species communities through cross-feeding. Building upon findings from their previous studies involving *S. cerevisiae*, the authors identified pairs of gene knockouts that promote cross-feeding and co-growth of auxotrophic mutants of *Y. lipolytica* and *S. cerevisiae*. These syntrophic co-cultures were characterized and harnessed for bioproduction purposes. The study showcased an enhanced production of 3-HP through a division of labor strategy involving *Yarrowia* and *Saccharomyces* mutants.

Specific comments

1. The authors presented only growth results in Figure 1 and Figure 2. It is suggested that sugar consumption profiles and (total) biomass yields from co-cultures be included to demonstrate enhanced substrate utilization by the co-cultures.

2. Unlike syntrophic co-cultures of *S. cerevisiae* in their previous publications and co-cultures of *Y. lipolytica* in Figures 1 and 2 of this manuscript, mixed cultures of *S. cerevisiae* (Crabtree positive) and *Y. lipolytica* (Crabtree-negative) need further examination under different levels of glucose and aeration. If growth profiles and cell densities of *S. cerevisiae* and *Y. lipolytica* change depending on glucose and aeration, the information obtained from Figure 3 might not be valid. Additional experiments are suggested to address this issue. Also, levels of byproducts (glycerol, acetate, and ethanol) need to be presented in Figure 3.

3. While the authors argue that the enhanced production of 3-HP by the syntrophic cocultures is due to a division of labor strategy, the results are not impressive enough to warrant the merit of the cocultures. The titers of 3-HP in the range of 4-5 mM (0.3-5 g/L) in Figure 4 are too low compared to published results using single-engineered yeast for producing 3-HP.

4. Again, glucose consumption and byproduct production profiles of the high-performing co-cultures in Figure 4 are suggested to be presented.

5. Levels of beta-alanine in the supernatant of co-cultures are suggested to be measured.

6. In Figure 1 (b), a legend for *dura3-dmet5* is missing.

7. In Figure 3 (d) and (e), legends for *trp2* and *trp4* are suggested to be added (as in Figure 2 d, e, f). Figure 3 (e) does not have the second axis title (OD600).

Version 1:

Reviewer comments:

Reviewer #1

(Remarks to the Author)

I am satisfied with the authors' response. The division of labour still remains an hypothesis though and it should be clearly stated as such.

Reviewer #2

(Remarks to the Author)

The authors have made appropriate revisions based on the reviewers' comments. This reviewer would like to suggest one additional point for clarification: In Supplementary Figure 10 (b) and (d), both ethanol and glycerol were detected in the cultures of YLdtrp2 and YLdtrp4. Given the small amounts of glucose consumption (~6 g/L) at the 1:10 inoculation ratio, the reported concentrations of ethanol (0.8 g/L), citrate (2.5 g/L), and glycerol (3 g/L) might be overestimated. Please verify whether both ethanol and glycerol were indeed produced and confirm the accuracy of these values.

RESPONSE TO REVIEWERS' COMMENTS

Reviewer #1 (Remarks to the Author):

The manuscript by Park et al shows that an engineered pathway split between different cells leads to increased production. The stability of the split system in terms of the co-existence is maintained by engineered obligate mutualism between the cell types (complementary auxotrophies). Interestingly, the engineered mutualism led to stable growth in intra- as well as inter-species systems consisting of two popular cell factories, *Saccharomyces cerevisiae* and/or *Yarrowia lipolytica*. The work is of interest and importance to the synthetic biology field and provides a proof-of-concept for using engineered communities for sustainable chemical production. However, the study in its current state is mostly observational and does not add much new insights beyond what has been shown before in *S. cerevisiae* communities (by the same lab; e.g. see Aulakh et al. Nature chemical biology 2023). Below are my main points and suggestions towards improving the study.

We appreciate the comments and suggestions to improve our manuscript. We have addressed all the comments and made additional experiments and several revisions, which we believe have improved the quality of the manuscript and its contribution to the field of synthetic microbial communities.

We have responded to each comment point-by-point below and we also summarise here some of the major changes:

- We have added quantification of the production of all metabolites in the 3-HP pathway, including pathway intermediates (β -alanine, malonic semialdehyde, and hydroxy propionic acid).
- In the discussion section and the supplementary information, we now compare the division of labor in co-cultures between our previous study and this study.
- We have expanded our results to add information on substrate consumption during the co-culture experiments.
- We have unified OD measurements throughout the manuscript for better readability.
- We have now presented insights into the potential nature of the division of labour, suggesting pyruvate availability as a key factor.
- We have performed additional experiments to evaluate the potential impact of the Crabtree effect on the co-cultures with *S. cerevisiae*.
- We have updated the main and supplementary figures and their corresponding raw data, which are in the supplementary tables.

As mentioned by the reviewer, our previous study studied syntrophic communities in the model yeast *S. cerevisiae* (Aulakh et al. 2023). We believe this manuscript goes one step further in the study of communities and highlight here some of the novel points of this study:

- In this work, the non-model yeast *Y. lipolytica* was engineered for syntrophic relationships, and its capacity to cross-feed metabolites was proved for the first time. This is, to our knowledge, the first report of synthetic syntrophy in a non-

model yeast. Given this yeast's increasing industrial relevance, the stable syntrophic synthetic communities described here pave the way for improved bioproduction.

- This study also proved, for the first time, spontaneous growth by cross-feeding metabolites between two yeast species. In the revised version, we also showed the different dynamics of the interspecies community in different culture conditions, paving the way for further studies, including higher-order combinations and different species.
- We also proved that division of labour was effective for bioproduction in interspecies syntrophic communities and showed that pyruvate availability could play a significant role. This suggests division of labour could be one solution to current limitations such as resource competition in highly engineered strains.
- These results are also relevant for the *Yarrowia* research community, which widely uses auxotrophies as selectable markers. From what we can see in this work, some of the widely used auxotrophic markers (Ura3 or Leu2) can be rescued by metabolite exchange, which suggests a risk of isolating colonies that are not prototrophs after a transformation, but selecting those that are growing out of the amino acids being exchanged by the correctly transformed cells. This can guide a better selection of selectable markers in the future, leading to more stable and efficient strain engineering.
- Division of labor was explored in a 3-step pathway (longer than in our previous work) in YL-YL and YL-SC syntrophic communities by using 3-hydroxypropionic acid as a proof-of-concept. This shows the potential of synthetic communities for bioproduction to improve longer and more complex pathways in the future.

1. There is neither conceptual distinction nor mechanistic insights compared to Aulakh et al. 2023. This could be addressed by a more detailed, molecular analysis of the co-cultures, using, e.g. proteomics. The main question is what drives higher production in the split system? If it is division of labour, what are the costs and where do they stem from? Or are there other biochemical reasons.

We appreciate this comment, which has made us look deeper into the potential mechanisms behind these co-cultures. We hypothesize that the higher production of 3-HP in the co-cultures compared to monoculture comes from a higher availability of the pathway precursor, pyruvate. This is because pyruvate is required on the one hand to make beta-alanine and on the other hand to take beta-alanine and make 3-HP. Division of labour could help redistribute the pyruvate pool, which would not need to be used to make both beta-alanine and 3-HP in a single cell (see pathway scheme in Supplementary Figure 16 (a) below).

Supplementary Figure 16. (a) Metabolic pathway including synthetic 3-HP pathway. (b) Production of metabolites (citrate, pyruvate, β -alanine, malonic semialdehyde, and 3-hydroxypropionic acid) from mono- and co-cultures. The strains were incubated in flask at 30 °C with 250 rpm for 120 hours. Values represent averages and error bars denote standard deviation (n=2). One-way ANOVA, followed by Bonferroni's multiple

comparisons test with 95% confidence intervals were performed using GraphPad Prism 9.5.0 software and p values are indicated as asterisks in the graph (*: $p < 0.05$, **: $p < 0.005$, ***: $p < 0.0005$, ****: $p < 0.0001$).

In addition, pyruvate is also the precursor of the byproduct, citrate, and a higher production of citrate could lead to a lower production of the desired product, as shown in Supplementary Figure 16 (a). Citrate is mainly produced during the stationary phase, as seen in the monoculture (Supplementary Figure 15 (a), and its production has been associated with cell growth limitation (de Veiga Moreira et al. 2021*; Carsanba et al. 2019**)). Therefore, the growth control provided by the synthetic syntrophic relationships and their associated slower dynamics of co-cultures could lead to lower citrate production.

Supplementary Figure 15. Profiles of glucose consumption and byproduct formation in the co-cultures for 3-HP production. The strains were incubated in flask at 30 °C with 250 rpm for 120 hours. Values represent averages and error bars denote standard deviation ($n=2$).

In order to support this hypothesis, we expanded our experimental analysis and quantified not only pathway intermediates but also pyruvate and citric acid. In the updated Supplementary Figure 15, we can observe how the monoculture rapidly reaches the stationary phase and produces citrate as a byproduct (also in Supplementary Figure 16 (b) and Additional Figure 1 below). Citrate is synthesized from pyruvate, which, as mentioned, is a precursor of 3-HP synthesis, potentially generating resource competition. From our metabolite analysis, we observed that the co-cultures showed slower growth and lower citrate formation compared to the YL monoculture (Supplementary Figure 15 (a) *versus* (b-f)), which resulted in higher 3-HP production with a concomitant accumulation of residual pyruvate (Supplementary Figure 16 (b) and Additional Figure 1 below). This suggests that the split of the 3-HP pathway might help provide more pyruvate to the reaction of BAPAT in both intra- and interspecies co-cultures, which could be a limiting factor in the monoculture.

*de Veiga Moreira et al. Fine-tuning mitochondrial activity in *Yarrowia lipolytica* for citrate overproduction. Scientific reports. 11 (2021)

** Carsanba et al. Citric acid production by *Yarrowia lipolytica*. In: Non-conventional Yeasts:from basic research to application. Springer, Cham. (2019)

Additional Figure 1. Comparison of key metabolites (pyruvate, citrate, and 3-HP) in the 3-HP pathway from the mono- and co-culture. The strains were incubated in flask at 30 °C with 250 rpm for 120 hours. Values represent averages and error bars denote standard deviation (n=2).

Future analysis based on C¹³-MFA could help us to confirm this hypothesis and give us a deeper understanding of how the metabolic pathways of the different populations in the syntrophic communities are regulated.

These new results are now added to the revised manuscript and the new figures cited in the text as shown below:

In the Result section: “*Instead, WT monoculture produced higher citrate than co-culture (Supplementary Fig. 15 and 16).*”

In the Discussion section: “*The growth and metabolite analysis (Supplementary Fig. 16) suggests that the higher 3-HP production found in the co-cultures originated from a higher availability of pyruvate, a common precursor of 3-HP and citrate.*”

2. The main difference to Aulakh et al 2023, regarding pathway, seems to be that in the present work, they have the complete pathway up to 3-hydroxypropionate whereas in Aulakh et al 2023, they only reported the addition of the BAPAT and PAND. However, in this work, they quantify the end product 3-hydroxypropionate instead of the intermediate malonic semialdehyde. There needs to be a detailed comparison with the Aulakh et al work in terms of observed yields, productivity etc and explanation of differences. For example, it seems that the final product was not quantified in the previous work while the pathway intermediates are not reported here.

We appreciate this comment. We have, accordingly, now added the production levels of the metabolites (β -alanine, malonic semialdehyde, and 3-hydroxypropionic acid) in the 3-HP pathway in Supplementary Table 4 and Supplementary Figure 12 -14.

Supplementary Table 4. Production of metabolites in the 3-HP biosynthetic pathway from the intra- and interspecies synthetic communities. Values represent average and standard deviation (n=2).

Strain(s)	Inoculation ratio	β -Alanine (mM)	MSA (mM)	3-HP (mM)
SC mono (P+B)	-	0.194 \pm 0.0259	0.051 \pm 0.0085	0.112 \pm 0.0175
YL mono (P+B)	-	0.071 \pm 0.0007	0.021 \pm 0.0017	0.242 \pm 0.0177
SC WT-P/SC WT-B	1:1	0.190 \pm 0.0021	0.061 \pm 0.0029	0.065 \pm 0.0047
YL WT-P/YL WT-B	1:1	0.092 \pm 0.0007	0.019 \pm 0.0005	0.113 \pm 0.0264
SC WT-B/YL WT-P	1:1	0.051 \pm 0.0027	0.038 \pm 0.0042	nd
SC Δ trp2-B/SC Δ trp4-P	10:1	nd	1.599 \pm 0.0267	0.104 \pm 0.0316
	1:1	nd	0.058 \pm 0.0142	0.14 \pm 0.0317
	1:10	nd	0.066 \pm 0.0066	nd
YL Δ trp2-B/YL Δ trp4-P	10:1	0.02 \pm 0.0196	1.089 \pm 0.0167	4.671 \pm 0.0635
	1:1	0.261 \pm 0.0347	1.761 \pm 0.0910	2.666 \pm 0.0473
	1:10	0.103 \pm 0.0026	0.039 \pm 0.0009	0.265 \pm 0.0355
SC Δ trp2-B/YL Δ trp4-P	10:1	nd	1.466 \pm 0.0637	4.500 \pm 0.0119
	1:1	nd	0.159 \pm 0.0210	0.204 \pm 0.0008
	1:10	0.088 \pm 0.0014	1.453 \pm 0.0029	0.076 \pm 0.0150
YL Δ trp2-B/SC Δ trp4-P	10:1	0.163 \pm 0.0030	1.473 \pm 0.0236	3.962 \pm 0.0009
	1:1	0.182 \pm 0.0032	0.163 \pm 0.0613	0.706 \pm 0.1052
	1:10	0.201 \pm 0.0213	0.950 \pm 0.1025	1.119 \pm 0.0477

*nd: not detected

The comparison of bioproduction between our two studies is shown in the new Supplementary Figure 18 included below. We would like to highlight that the co-cultures in the previous study and this study were done with different strains, expressing genes under different promoters (see answers to comment #4) and different cultivation scales (48-well deep plate vs. flask). While this makes the comparison difficult, as proxy, we compared the trend of production of the common metabolites (either MSA only or the

sum of MSA and 3-HP- which would represent the products from the second member of community) rather than the exact production level.

Supplementary Figure 18. Comparison of division of labor in the *S. cerevisiae* synthetic community between previous study and this study. Supplementary Table 10 of previous study (Aulakh et al. 2013) was used.

We also included the comparison on MSA production from these two studies in the discussion.

“When it comes to MSA production in *S. cerevisiae* communities, the coculture of Δ trp2-B: Δ trp4-P=10:1 produced the highest MSA among different inoculation ratios but also outperformed the monoculture, which is consistent with the MSA production in a previous study of *S. cerevisiae* communities (Supplementary Table 4 and Supplementary Figure 14 and 18) [Aulakh et al. 2023]. However, the production of MSA in *S. cerevisiae* co-culture at Δ trp2-B: Δ trp4-P=1:1 and 1:10 was negligible, although it was higher than the monoculture in the previous study. This might be due to the different promoters used for expressing BAPAT in each study, additional gene expression (YDFG) in this study, and different cultivation scales.

In the case of 3-HP production, *S. cerevisiae* co-culture at specific inoculation ratio (Δ trp2-B: Δ trp4-P=1:1) performed better than the *S. cerevisiae* monoculture (Supplementary Figure 14, Supplementary Table 4). The level of total metabolites produced from the Δ trp2-B strain (MSA and 3-HP) in the coculture of Δ trp2-B: Δ trp4-P=10:1 is higher than the one from the monoculture (Supplementary Table 4 and Supplementary Figure 14).”

3. With regards to the enhancement of bioproduction, the authors use the full 3-HP pathway rather than the same pathway up to malonic semialdehyde as they reported in Aulakh et al 2023. It is unclear why they chose different lengths of the same pathway for these two works as the auxotroph pairs and the experimental design are otherwise almost identical.

In our previous study, we worked on MSA production, which was a more direct measurement for validating the division of labor, as it involves only a two-step pathway. Thus, two enzymes (PAND, BAPAT) were individually expressed in each strain for the coculture.

In this study, we wanted to evaluate the division of labor in a longer pathway to produce 3-HP, which is a more stable product that has also gained interest as a building block in chemistry industry. The division of labor has been proved to be effective for splitting a different three-step pathway into two members community in another study [Peng et al. 2024]. While we applied the same upstream pathway until MSA as in our previous study, the expression system (promoter and terminator), the species, and the cultivation scale applied to this study were different. We would like to highlight that the 3-HP pathway was selected as a proof-of-concept to validate division of labor in our newly established intra- and interspecies syntrophic communities rather than follow and compare with previous studies. As mentioned in the answer to comment #2, we have now included the comparison of results between these two studies in the discussion.

Their results indicate that when 3-HP is used instead of MSA to quantify bioproduction efficiency of this synthetic pathway, *Yarrowia lipolytica* co-cultures and *Y. lipolytica* - *S. cerevisiae* cocultures perform better than the *Y. lipolytica* monoculture while the *S. cerevisiae* monoculture performs better than the *S. cerevisiae* co-cultures. This contradicts the results of malonic semialdehyde production reported in their previous publication (Aulakh 2023), however, the authors do not discuss why this might be and how these results may be reconciled.

As discussed in the previous answers, we have now added the quantification of the production of metabolites of the 3-HP pathway (β -alanine, malonic semialdehyde, and 3-hydroxypropionic acid) in Supplementary Table 4 (as presented in the response to comment #2) and Supplementary Figure 12 - 14.

As added in the discussion (answered to comment #2), the *S. cerevisiae* co-culture at a certain inoculation ratio ($\Delta trp2-B:\Delta trp4-P=10:1$) performed better for MSA production compared to the SC monoculture, which is consistent with the previous study. This effect was more significant when we compared the production of metabolites (MSA + 3-HP) in the co-culture. The co-culture $\Delta trp2-B:\Delta trp4-P=10:1$ outperformed by 10-fold the monoculture (1.703 mM versus 0.163 mM) as shown in the Additional Table 1 included below and Supplementary Figure 18 (c). Together with the analysis of pyruvate and citrate, as answered to comment #1, we suggest that the division of labor help redistribute the precursor pool, thus improving the production of 3-HP in the co-culture.

Additional Table 1. Production of metabolites in the 3-HP biosynthetic pathway from the SC mono- and co-cultures. Values represent average and standard deviation (n=2).

Strain(s)	Inoculation ratio	β -Alanine (mM)	MSA (mM)	3-HP (mM)	MSA + 3-HP (mM)
SC mono (P+B)	-	0.194 \pm 0.0259	0.051 \pm 0.0085	0.112 \pm 0.0175	0.163
SC WT-P/ SC WT-B	1:1	0.190 \pm 0.0021	0.061 \pm 0.0029	0.065 \pm 0.0047	0.126
SC Δ trp2-B/ SC Δ trp4-P	10:1	nd	1.599 \pm 0.0267	0.104 \pm 0.0316	1.703
	1:1	nd	0.058 \pm 0.0142	0.14 \pm 0.0317	0.199
	1:10	nd	0.066 \pm 0.0066	nd	0.066

Supplementary Figure 18. Comparison of division of labor in the *S. cerevisiae* synthetic community between previous study and this study. Supplementary Data 8 of previous study (Aulakh et al. 2013) was used.

4. The authors state that the *S. cerevisiae* strains used in the present study bear the complete 3-HP pathway as reported in Borodina et al 2015 and give Aulakh 2023 as the reference for the construction of this strain. However, in Aulakh 2023, they reported only the insertion of BcBAPAT and TcPAND in their strains. For example, in Aulakh et al 2023, Supplementary Table 10, EcYDFG is not mentioned as being present in any strain. However, in Table 1 of the present work, the authors report that strain “RLA S1629” contains EcYDFG and the source as Aulakh et al 2023. It is therefore unclear whether the authors used the complete 3-HP pathway in their previous work (Aulakh 2023) and did not report the presence of the EcYDFG gene being inserted into the strains they created in that work or if the authors created new strains for the present work which bear EcYDFG in addition to the BcBAPAT and TcPAND.

We appreciate the reviewer for noticing this. We have now corrected the information on plasmids and strains with the correct reference citation in Table 1. The plasmids and strains used in this study were newly constructed with different promoters (pTDH3, pCCW12, and pPGK1) and terminators (tENO1, tSSA1, tADH1) by using YTK Golden Gate system. Each gene (*PAND*, *BAPAT*, and *YDFG*) in the 3-HP synthetic pathway was expressed under different promoters while the genes (*PAND* and *BAPAT*) in the Aulakh et al. 2023 strains used the same promoter (pTDH3) and the same terminator (tADH1) as shown in the Supplementary Table 10 included below.

[Corrected parts in Table 1 in this manuscript]

Plasmids		Reference
RLA P1676	pTDH3-TcPAND-tENO1-vLeu2	This study
RLA P1679	pCCW12-BcBAPAT-tSSA1- pPGK1-EcYDFG-tADH1-vLeu2	This study
RLA P1683	pTDH3-TcPAND-tENO1- pCCW12-BcBAPAT-tSSA1- pPGK1-EcYDFG-tADH1-vLeu2	This study
S. cerevisiae		
RLA S1628	BY4741 pTDH3-TcPAND-tENO1+ pHUM	This study
RLA S1629	BY4741 pCCW12-BcBAPAT-tSSA1- pPGK1-EcYDFG-tADH1-tLeu2 + pHUM + LEU2	This study
RLA S1585	BY4741 pTDH3-TcPAND-tENO1- pCCW12-BcBAPAT-tSSA1, pPGK1-EcYDFG-tADH1-tLeu2, pHUM + pHUM + LEU2	This study
RLA S1604	BY4741 Δ trp2 + pTDH3-TcPAND-tENO1 + pHM + LYS21	This study
RLA S1605	BY4741 Δ trp2 pCCW12-BcBAPAT-tSSA1- pPGK1-EcYDFG-tADH1-tLeu2 + pHM + LYS21	This study
RLA S875	BY4741 Δ TRP4 + pTDH3-TcPAND-tENO1 + pHLM	This study
RLA S894	BY4741 Δ TRP4 + pCCW12-BcBAPAT-tSSA1- pPGK1-EcYDFG-tADH1-tLeu2 + pHLM	This study

[Part of Supplementary Table 10. From Aulakh et al. Nat Chem Bio 2023]

trp2Δ-TcPAND	trp2Δ , MATa, his3 Δ , leu2 Δ , met15 Δ , ura3 Δ	BY4741	pTDH3-TcPAND-tADH1 + pHLM	RLA laboratory
trp2Δ-BcBAPAT	trp2Δ , MATa, his3 Δ , leu2 Δ , met15 Δ , ura3 Δ	BY4741	pTDH3-BcBAPAT -tADH1+ pHLM	RLA laboratory

5. There is a 100-fold discrepancy between the OD600 values reported in various figures. For example, OD600 range in Figure 1 is up to 0.8, but the OD600 maximum in Figure 4 (c) is 60. Is this solely due to the difference in growth between 96-well cultures and flasks or have two different instruments been used to measure the optical density without calibration to a single axis? How these OD measurements relate to cell numbers? Is the cell density in flasks 75-times that in 96 well plates?

We appreciate the comment. The main difference in OD₆₀₀ between Figure 1 and Figure 4 came from the different cultivation scales (96 well plates and flask) described in the Materials and Methods and the different instruments used to measure OD.

Following the comment of the reviewer, we have now normalized the OD value across all our results after calibrating to the one from the microplate reader (Figure 4 (c), Supplementary Figure 12-14). For that, the calibration curve was used to normalize OD₆₀₀ values from the spectrophotometer and microplate reader, as shown below and added in Supplementary Figure 17. This calculation is described in the Materials and Methods. For clarity and traceability, we also included raw OD₆₀₀ data and normalized OD₆₀₀ data in Supplementary Table 5.

Supplementary Figure 17. Calibration curve of OD₆₀₀ between microplate reader and spectrophotometer. The OD₆₀₀ from spectrophotometer was calibrated to the value of microplate reader in figures in the main manuscript and Supplementary Figures.

6. Based on the growth curve in Figure 4c, it seems that the readings for most of the exponential phase are missing.

For the experiment in Figure 4c, we measured the OD₆₀₀ every 24 hours, which was sufficient to show us some general trends about the growth patterns between WT cultures (mono and coculture) and auxotroph cultures (cocultures) and the maximum biomass production. That was the goal of this experiment rather than capturing the details of the exponential phase. The WT cultures showed the exponential phase between 0 – 24 hours, while cocultures started the exponential growth after 24 hours (between 24 – 48 hours), being especially slow for YL Δ trp2-B/YL Δ trp4-P=10:1.

7. The results of the metabolite quantification via LC-MS should be provided in a table form in the supplementary data. The mass-spectrometry analysis methods are unclear – are these targeted or untargeted measurements? From the methods it seems like untargeted analysis, which is odd given that only 3 metabolites are reported. Were standards used to validate these? The authors should also submit the raw mass spectrometry data to a community repository as is custom for mass spectrometry experiments.

Metabolite analysis via LC-MS was a targeted measurement with the standards (β -alanine and 3-hydroxypropionic acid). In the case of malonic semialdehyde, it was semi-quantified by the standard curves of β -alanine due to the failure in getting it synthesized by the chemical companies we attempted. This is now described with standard information in Materials and Methods as indicated below. Raw mass spectrometry data is now uploaded in the supplementary Table 6.

“Metabolites in the 3-HP pathway, β -alanine, malonic semialdehyde, and 3-HP, were analyzed by LC-MS [...] All metabolites were qualified by the functional m/z values. β -Alanine and 3-HP were quantified by the calibration curve of the standards. Malonic semialdehyde was semi-quantified by the standard curves of β -alanine (Aulakh et al. 2023).”

8. Since, the authors do not provide tables for their bioproduction capacity metabolite quantifications, one can only infer from the Line 261 “4.50mM which is 40.3 times higher than WT *S. cerevisiae* 18.6 times higher than WT *Y. lipolytica*” that the titre in WT Sc is ~0.11mM and WT Yl is ~0.24mM. This suggests that Yl is a better host for the 3-HP production than Sc. Why is this?

Thanks for the valuable comment. A new table including the production result of 3-HP (and other pathway intermediates) has now been added as supplementary Table 4.

Strain(s)	Inoculation ratio	b-Alanine (mM)	MSA (mM)	3-HP (mM)
SC mono (P+B)		0.194 ± 0.0259	0.051 ± 0.0085	0.112 ± 0.0175
YL mono (P+B)		0.071 ± 0.0007	0.021 ± 0.0017	0.242 ± 0.0177
SC WT-P/SC WT-B	1:1	0.190 ± 0.0021	0.061 ± 0.0029	0.065 ± 0.0047
YL WT-P/YL WT-B	1:1	0.092 ± 0.0007	0.019 ± 0.0005	0.113 ± 0.0264
SC WT-B/YL WT-P	1:1	0.051 ± 0.0027	0.038 ± 0.0042	nd
SC Δ trp2-B/SC Δ trp4-P	10:1	nd	1.599 ± 0.0267	0.104 ± 0.0316
	1:1	nd	0.058 ± 0.0142	0.14 ± 0.0317
	1:10	nd	0.066 ± 0.0066	nd
YL Δ trp2-B/YL Δ trp4-P	10:1	0.02 ± 0.0196	1.089 ± 0.0167	4.671 ± 0.0635
	1:1	0.261 ± 0.0347	1.761 ± 0.0910	2.666 ± 0.0473
	1:10	0.103 ± 0.0026	0.039 ± 0.0009	0.265 ± 0.0355
SC Δ trp2-B/YL Δ trp4-P	10:1	nd	1.466 ± 0.0637	4.500 ± 0.0119
	1:1	nd	0.159 ± 0.0210	0.204 ± 0.0008
	1:10	0.088 ± 0.0014	1.453 ± 0.0029	0.076 ± 0.0150
YL Δ trp2-B/SC Δ trp4-P	10:1	0.163 ± 0.0030	1.473 ± 0.0236	3.962 ± 0.0009
	1:1	0.182 ± 0.0032	0.163 ± 0.0613	0.706 ± 0.1052
	1:10	0.201 ± 0.0213	0.950 ± 0.1025	1.119 ± 0.0477

*nd: not detected

As suggested by the reviewer, the *Y. lipolytica* monoculture seems to be a better producer than the *S. cerevisiae* one. Looking at the accumulation of intermediates, it seems that the metabolic flux through the pathway is more optimal in *Y. lipolytica*, where there was a lower accumulation of beta-alanine and malonic semialdehyde. In accordance, the co-culture without auxotrophies YL-YL showed better 3HP production than the SC-SC one, where the latter also accumulated a higher amount of intermediates (beta-alanine and malonic semialdehyde).

However, it is worth noting that the 3-HP production from YL-SC co-culture (4.50 mM) was similar to the one from YL-YL co-culture (4.67 mM), being those two the highest producers.

9. Minor issues:

Most figures do not show actual data points, which should be included in all cases

We included the actual data points as dots in the bar graphs in Fig 1 (a), Fig 2 (d-i), Fig 3 (a), (d), (e), and Fig 4 (d) in the original manuscript, but it might not be very visible. We modified the graph to show each data point better. In the case of the growth curves, we kept the graph as they are and added the raw data in the revised manuscript as Supplementary Table 7.

Figure 3a : why is there a > sign above the monocultures ?
p was missing in the figure, we now modified it with “p>”.

Line 76 - community should be communities
“could be originated by the needs of each population”
We corrected this.

149 - “tryptophane”- extra “e”
We corrected this.

Reviewer #2 (Remarks to the Author):

This manuscript by Park et al. elucidates the genetic backgrounds of yeast strains, specifically *Yarrowia lipolytica* and *Saccharomyces cerevisiae*, facilitating both intra- and inter-species communities through cross-feeding. Building upon findings from their previous studies involving *S. cerevisiae*, the authors identified pairs of gene knockouts that promote cross-feeding and co-growth of auxotrophic mutants of *Y. lipolytica* and *S. cerevisiae*. These syntrophic co-cultures were characterized and harnessed for bioproduction purposes. The study showcased an enhanced production of 3-HP through a division of labor strategy involving *Yarrowia* and *Saccharomyces* mutants.

We appreciate the comments and suggestions on our manuscript. We have addressed all the comments and made several major revisions which have improved the quality of this study. Please find below point-by-point responses to your comments.

Specific comments

1. The authors presented only growth results in Figure 1 and Figure 2. It is suggested that sugar consumption profiles and (total) biomass yields from co-cultures be included to demonstrate enhanced substrate utilization by the co-cultures.

We appreciate this comment. As suggested, we evaluated the glucose consumption in the mono- and co-culture described in Figures 1 and 2. This new information is now added as Supplementary Figures 3 and 4 for Figures 1 and 2 in the main manuscript, respectively. From the result, we could observe a positive correlation between syntrophic growth and glucose consumption depending on the auxotroph pairs (eg. *Δura3-Δtrp4*, *Δtrp2-Δtrp4*, and *Δtrp4-Δmet5*) and inoculation ratios. Auxotroph strains in monoculture which showed no growth, resulted in no glucose consumption. Glucose consumption was different depending on the inoculation ratios (Supplementary Figure 4), which correspond to the growth trend. It is worth noting that the tests were done in 96 well plates; thus, the consumption of glucose is limited compared to the culture in flasks, as seen in Supplementary Figure 15.

We also would like to highlight that the objective of our study is to establish the syntrophic interaction between auxotroph strains that exchange essential metabolites with each other rather than enhancing substrate consumption.

These results are now included in the manuscript (line 109-111, line 144-145).

Supplementary Figure 3. Glucose consumption of monocultures (a single auxotrophe) and cocultures (a pair of two auxotroph with 1:1 inoculation ratio) at 72 hours of cultivation. Values represent averages and error bars denote standard deviation (n=3). One-way ANOVA, followed by Bonferroni's multiple comparisons test with 95% confidence intervals were performed using GraphPad Prism 9.5.0 software and *p* values are indicated as asterisks in the graph (*:*p* < 0.05, **: *p* < 0.005, ***: *p* < 0.0005, ****: *p* < 0.0001).

Supplementary Figure 4. Glucose consumption of selected syntrophic coculture of *Y. lipolytica* auxotroph strains at different inoculation ratio from 10:1 to 1:10. Values represent averages and error bars denote standard deviation (n=3).

2. Unlike syntrophic co-cultures of *S. cerevisiae* in their previous publications and co-cultures of *Y. lipolytica* in Figures 1 and 2 of this manuscript, mixed cultures of *S. cerevisiae* (Crabtree positive) and *Y. lipolytica* (Crabtree-negative) need further examination under different levels of glucose and aeration. If growth profiles and cell densities of *S. cerevisiae* and *Y. lipolytica* change depending on glucose and aeration, the information obtained from Figure 3 might not be valid. Additional experiments are suggested to address this issue. Also, levels of byproducts (glycerol, acetate, and ethanol) need to be presented in Figure 3.

This is a really great comment. Accordingly, we have performed a set of new experiments. First, we have now added the metabolite profiles (glucose, ethanol, citrate, and glycerol) of Figure 3 (strain pair, glucose concentration, and inoculation ratio) as Supplementary Figure 10. The glucose consumption was positively correlated with the growth in each strain pair and varied with the inoculation ratio. The YL-YL co-culture produced more citrate and less ethanol than the ones from SC-YL co-culture.

Supplementary Figure 10. Profiles of metabolites in co-culture among $Y\Delta trp2$, $Y\Delta trp4$, $SC\Delta trp2$, and $SC\Delta trp4$ at different inoculation ratios in aerobic conditions. Metabolites (glucose, ethanol, citrate, and glycerol) from the co-culture of (a-d) $Y\Delta trp2$ - $Y\Delta trp4$ and (e-h) $SC\Delta trp2$ - $Y\Delta trp4$. Values represent averages and error bars denote standard deviation (n=3).

Second, in order to verify if the syntrophic growth is affected by the Crabtree effect of *S. cerevisiae*, and as suggested, we performed the co-culture of YL-YL and SC-YL with different glucose concentrations (20 and 100 g/L) and aeration conditions (aerobic and semi-anaerobic) (Supplementary Figure 11). In general, for the coculture $\Delta trp2$: $\Delta trp4$, we observed a higher growth at the inoculation ratio of 1:10 than at 10:1 and glucose consumption varied depending on their growth.

At 20 g/L of glucose, $SC\Delta trp2 : YL\Delta trp4 = 1:1$ grew similarly to $SC\Delta trp2 : YL\Delta trp4 = 1:10$ in semi-anaerobic conditions, while no growth was observed in aerobic conditions. We observed the higher production of ethanol in this condition ($SC\Delta trp2 : YL\Delta trp4 = 1:1$, 20g/L of glucose, semi-anaerobic), suggesting that the Crabtree effect helped $SC\Delta trp2$ strain grow in this condition.

At a higher glucose concentration (100 g/L), in both aerobic and semi-anaerobic conditions, we generally observed higher growth when there was a higher presence of the $\Delta trp4$ strain. In the SC-YL co-culture, we observed the Crabtree effect, as reflected by the ethanol production (100 g/L glucose, semi-anaerobic condition) that seemed to come from $SC\Delta trp2$. As expected, a negligible amount of ethanol was observed in YL-YL co-culture in the same condition.

These results support the comment of the reviewer and confirm that syntrophic growth can be varied depending on the conditions. Thus, we have now added this information to the main manuscript (line 192-205).

Supplementary Figure 11. Profiles of growth and metabolites in co-culture among $YL\Delta trp2$, $YL\Delta trp4$, $SC\Delta trp2$, and $SC\Delta trp4$ at different inoculation ratios in different culture condition. (a) Aerobic condition (continuous shaking), and (b) semi-anaerobic condition (static and closed condition). Values represent averages and error bars denote standard deviation (n=3).

(a)

Glucose	20 g/L	100 g/L	20 g/L	100 g/L
Co-culture	$YL\Delta trp2$ - $YL\Delta trp4$	$YL\Delta trp2$ - $YL\Delta trp4$	$SC\Delta trp2$ - $YL\Delta trp4$	$SC\Delta trp2$ - $YL\Delta trp4$
Aeration	aerobic	aerobic	aerobic	aerobic

(b)

Glucose	20 g/L	100 g/L	20 g/L	100 g/L
Co-culture	YLΔtrp2-YLΔtrp4	YLΔtrp2-YLΔtrp4	SCΔtrp2-YLΔtrp4	SCΔtrp2-YLΔtrp4
Aeration	Semi-anaerobic	Semi-anaerobic	Semi-anaerobic	Semi-anaerobic

3. While the authors argue that the enhanced production of 3-HP by the syntrophic cocultures is due to a division of labor strategy, the results are not impressive enough to warrant the merit of the cocultures. The titers of 3-HP in the range of 4-5 mM (0.3-5 g/L) in Figure 4 are too low compared to published results using single-engineered yeast for producing 3-HP.

We agree with the reviewer, however we would like to highlight that the objective of this study is to present a proof of concept for division of labor rather than to produce the highest amount of 3-HP. In addition, our production levels are comparable to those of previously published starting strains (with similar level of modifications) as described in the discussion. The expression of a single copy of the biosynthetic pathway resulted in 0.35 g/L in *Y. lipolytica* (Liu et al. 2023*) and 0.83 g/L in *S. cerevisiae* (Borodina et al. 2015**).

In the mentioned studies, further strain engineering and fermentation optimization led to a significant increase in production, 16.23 g/L in *Y. lipolytica* and 13.7 g/L in *S. cerevisiae*. This suggests the potential to increase the production of 3-HP in the coculture system after additional metabolic engineering and process optimisation, which is out of the scope of the current work but is of great interest for future research.

[Comparison of 3-HP production depending on the species and strategies]

Host	3-HP production	Strategy	Scale	Reference
Y. lipolytica – Y. lipolytica	0.42 g/L	Division of labor by co-culture (PAND, BAPAT, YDFG)	Flask	This study

Y. lipolytica – S. cerevisiae	0.405 g/L	Division of labor by co-culture (PAND, BAPAT, YDFG)	Flask	This study
Y. lipolytica	0.35 g/L	Introducing 3-HP biosynthetic pathway (MCRC, MCRN)	Flask	Lui et al. 2023*
	1.128 g/L	Further engineering (GAPNSm , ACC1 , ACSL641P , Δ MLS1 Δ CIT2)	Flask	
	16.23 g/L	Glucose feeding	Fed-batch fermentation	
S. cerevisiae	0.83 g/L	Introducing 3-HP biosynthetic pathway (PAND, BAPAT, YDFG)	Flask	Borodina et al. 2015**
	1.27 g/L	Further engineering (multicopy of PAND, PYC1 , PYC2 , AAT2)	Flask	
	13.7 g/L	Glucose limited feeding	Fed-batch fermentation	

*Lui et al. Engineering 3-Hydroxypropionic Acid Production from Glucose in *Yarrowia lipolytica* through Malonyl-CoA Pathway. *J. Fungi* **9**, (2023)

Borodina et al. establishing a synthetic pathway for high-level production of 3-hydroxypropionic acid in *Saccharomyces cerevisiae* via β -alanine. *Metab. Eng.* **27, 57–64 (2015).

4. Again, glucose consumption and byproduct production profiles of the high-performing co-cultures in Figure 4 are suggested to be presented.

We appreciate this comment. As suggested, we have performed additional experiments and the profiles of glucose consumption and byproduct production in selected co-cultures are now included in Supplementary Figure 15, which has been included below.

From the mono- and co-culture of *Y. lipolytica* we observed a production of citrate as a byproduct, while from the co-culture of *S. cerevisiae* and *Y. lipolytica* we found both citrate and ethanol, which was also observed in Supplementary Figure 10 (response to comment #2). The monoculture of *Y. lipolytica* reached the stationary phase rapidly and produced the highest level of citrate, while the *Y. lipolytica* co-cultures showed relatively slower growth and lower citrate production but higher 3-HP production, especially with the inoculation ratio of 10:1 of YL Δ trp2-B/YL Δ trp4-P.

Supplementary Figure 15. Profiles of glucose consumption and byproduct formation in the co-cultures for 3-HP production. The strains were incubated in flask at 30 °C with 250 rpm for 120 hours. Values represent averages and error bars denote standard deviation (n=2).

5. Levels of beta-alanine in the supernatant of co-cultures are suggested to be measured.

Thanks for this comment. Accordingly, the metabolites in the 3-HP pathway (β -alanine, malonic semialdehyde, and 3-HP) have now been included in Supplementary Figure 12 - 14. We have also included the production of metabolites in the 3-HP biosynthetic pathway in Supplementary Table 4 for better comparison and analysis, as shown below.

Supplementary Table 4. Production of metabolites in the 3-HP biosynthetic pathway from the intra- and interspecies synthetic communities. Values represent average and standard deviation (n=2).

Strain(s)	Inoculation ratio	β -Alanine (mM)	MSA (mM)	3-HP (mM)
SC mono (P+B)		0.194 \pm 0.0259	0.051 \pm 0.0085	0.112 \pm 0.0175
YL mono (P+B)		0.071 \pm 0.0007	0.021 \pm 0.0017	0.242 \pm 0.0177
SC WT-P/SC WT-B	1:1	0.190 \pm 0.0021	0.061 \pm 0.0029	0.065 \pm 0.0047
YL WT-P/YL WT-B	1:1	0.092 \pm 0.0007	0.019 \pm 0.0005	0.113 \pm 0.0264
SC WT-B/YL WT-P	1:1	0.051 \pm 0.0027	0.038 \pm 0.0042	nd
SC Δ trp2-B/SC Δ trp4-P	10:1	nd	1.599 \pm 0.0267	0.104 \pm 0.0316
	1:1	nd	0.058 \pm 0.0142	0.14 \pm 0.0317
	1:10	nd	0.066 \pm 0.0066	nd
YL Δ trp2-B/YL Δ trp4-P	10:1	0.02 \pm 0.0196	1.089 \pm 0.0167	4.671 \pm 0.0635
	1:1	0.261 \pm 0.0347	1.761 \pm 0.0910	2.666 \pm 0.0473
	1:10	0.103 \pm 0.0026	0.039 \pm 0.0009	0.265 \pm 0.0355
SC Δ trp2-B/YL Δ trp4-P	10:1	nd	1.466 \pm 0.0637	4.500 \pm 0.0119
	1:1	nd	0.159 \pm 0.0210	0.204 \pm 0.0008
	1:10	0.088 \pm 0.0014	1.453 \pm 0.0029	0.076 \pm 0.0150
YL Δ trp2-B/SC Δ trp4-P	10:1	0.163 \pm 0.0030	1.473 \pm 0.0236	3.962 \pm 0.0009
	1:1	0.182 \pm 0.0032	0.163 \pm 0.0613	0.706 \pm 0.1052
	1:10	0.201 \pm 0.0213	0.950 \pm 0.1025	1.119 \pm 0.0477

*nd: not detected

6. In Figure 1 (b), a legend for dura3-dmet5 is missing.

We corrected the legend of Figure 1 (b).

7. In Figure 3 (d) and (e), legends for trp2 and trp4 are suggested to be added (as in Figure 2 d, e, f). Figure 3 (e) does not have the second axis title (OD600).

We correct the legends and the axis title.

REVIEWERS' COMMENTS

Reviewer #1 (Remarks to the Author):

I am satisfied with the authors' response. The division of labour still remains an hypothesis though and it should be clearly stated as such.

We appreciate the comment and suggestion. Though the pathway split of 3-hydroxypropionic acid (3 component-pathway) showed improved production in this study, we agree with the reviewer that we still need more studies to really explain the mechanisms behind division of labour. Therefore, we have changed the text to clarify this.

“pathway split (and likely division of labor)”

“It is worth noting that further research is required to understand the complex relationship between division of labour and bioproduction and fully correlate them both.”

Reviewer #2 (Remarks to the Author):

The authors have made appropriate revisions based on the reviewers' comments. This reviewer would like to suggest one additional point for clarification: In Supplementary Figure 10 (b) and (d), both ethanol and glycerol were detected in the cultures of YLdtrp2 and YLdtrp4. Given the small amounts of glucose consumption (~6 g/L) at the 1:10 inoculation ratio, the reported concentrations of ethanol (0.8 g/L), citrate (2.5 g/L), and glycerol (3 g/L) might be overestimated. Please verify whether both ethanol and glycerol were indeed produced and confirm the accuracy of these values.

We appreciate this comment. After re-verification of HPLC raw data, we found out the error on glycerol data of YL Δ trp2- YL Δ trp4 co-culture. At 24 hours of culture, no glycerol was detected. This is now revised in Supplementary Figure 10 (d) as below.